# THE IMPACT OF ALLOCATION STRATEGIES IN SUBSET LEARNING ON THE EXPRESSIVE POWER OF NEURAL NETWORKS

**Ofir Schlisselberg**
Tel Aviv University ofirs4@mail.tau.ac.il

**Ran Darshan**
Tel Aviv University darshan@tauex.tau.ac.il

## ABSTRACT

In traditional machine learning, models are defined by a set of parameters, which are optimized to perform specific tasks. In neural networks, these parameters correspond to the synaptic weights. However, in reality, it is often infeasible to control or update all weights. This challenge is not limited to artificial networks but extends to biological networks, such as the brain, where the extent of distributed synaptic weight modification during learning remains unclear. Motivated by these insights, we theoretically investigate how different allocations of a fixed number of learnable weights influence the capacity of neural networks. Using a teacher-student setup, we introduce a benchmark to quantify the expressivity associated with each allocation. We establish conditions under which allocations have 'maximal' or 'minimal' expressive power in linear recurrent neural networks and linear multi-layer feedforward networks. For suboptimal allocations, we propose heuristic principles to estimate their expressivity. These principles extend to shallow ReLU networks as well. Finally, we validate our theoretical findings with empirical experiments. Our results emphasize the critical role of strategically distributing learnable weights across the network, showing that a more widespread allocation generally enhances the network's expressive power.

## 1 INTRODUCTION

A foundational principle in neuroscience posits that changes in synaptic weights drive learning and adaptive behaviors (Martin et al., 2000; Humeau and Choquet, 2019). This principle is mirrored in artificial neural networks (NNs), where modern algorithms adjust weights when training a network to perform a task. However, while typically in NNs it is common for all weights to be adaptable, the scale of this process in the brain is unclear.

Recent evidence suggests that only a small subset of synaptic weights is modified when an animal learns a new task (Hayashi-Takagi et al., 2015) and that training a subset of neurons can induce broad changes in neural activity as animals acquire new skills (Kim et al., 2023). Perturbing just a few neurons has been shown to significantly alter decision-making, perception, and memory-guided behaviors (Daie et al., 2021; Marshel et al., 2019; Robinson et al., 2020). These findings raise fundamental questions about the distributed nature of learnable weights in intelligent systems: To what extent are synaptic weight changes spread throughout the network, and what strategies should the learnable system use to allocate the subset of learnable weights?

While most algorithms used in NNs do not constrain which weights are trained, a few research directions explore this question, primarily from a practical standpoint. For example, training only a subset of the weights is used for pruning networks to make models more suitable for storage-constrained environments (Guo et al., 2021), to reduce computational costs (Thangarasa et al., 2023) and to reduce communication overheads in distributed systems (Sung et al., 2021). Similarly, transfer learning often fine-tunes large models by adjusting only a fraction of the weights ("parameter

allocating," see Wang et al. (2024)). Such a strategy is particularly useful for continual learning, helping to mitigate catastrophic forgetting (e.g., Mallya and Lazebnik, 2018; Mallya et al., 2018; Serrà et al., 2018; Wortsman et al., 2020; Zaken et al., 2022).

In both biological and artificial neural network research, similar questions arise regarding learning with only a subset of the available weights: If resources are constrained, what are the most effective strategies to allocate the learnable weights? Should learnable weights be confined to specific subsets of neurons, or distributed more broadly? And given an allocation strategy, how well can a network perform a task? Motivated by these questions, we theoretically study how learnable weights should be allocated within a network. More generally, we consider a model in which a task is learned under resource constraints—where only a fraction of the model's parameters is adaptable, while others remain fixed. In this setting, we explore how the selection of learnable parameters affects overall performance.

## 1.1 Our contribution

In this paper, we provide the first theoretical framework for analyzing the expressive power of various allocation strategies in NNs. Motivated by our goal of understanding how learnable weights should be organized in the brain, we apply our framework to explore the impact of allocating learnable weights on the expressivity of recurrent neural networks (RNNs). RNNs are particularly relevant to neuroscience as they serve as models to how neural systems maintain and process information over time (Hopfield, 1982; Elman, 1990; Barak, 2017; Qian et al., 2024). We focus on linear RNNs (LRNNs) due to their analytical tractability, grounded in the well-established literature on linear dynamical systems (e.g Heij et al. (2006)). Notably, we found that subset learning in LRNNs often produces non-trivial results, with insights that extend to feedforward architectures and even shallow ReLU networks. Our specific contributions are:

- We formalize the problem of how to allocate learnable parameters in a model using a student-teacher setup. We introduce a benchmark (Definition 2.3), which defines the match probability—the likelihood that a student, with a specific allocation of learnable parameters, can replicate the teacher's outputs. This measure of expressivity allows us to determine which allocations maximize the model's expressive power.

- For LRNNs, we prove several theorems that highlight the effects of different allocation strategies for learnable parameters in the encoder, decoder, and in the recurrent interactions (Theorems 3.1, 3.2, 3.4 and 3.5). These theorems identify conditions under which allocations can be maximal, leading to full expressivity, or minimal, resulting in zero expressive power. For cases where neither conditions are met, we propose heuristic principles to estimate the match probability. These results show a sharp transition between allocations with minimal expressivity to maximal expressivity.

- We show that similar concepts from LRNNs apply for fully connected multi-layer linear feed-forward networks (LFFN). We use these concepts to provide similar conditions that identify allocations that lead to large match probability.

- We show that similar concepts can be used to analyze the performance of the possible allocation strategies in one-layer ReLU feed-forward network.

Our theoretical findings suggest that, as a rule of thumb, allocations tend to become more optimal when distributing the learnable weights throughout the network. Specifically, distributing the *same* number of learnable weights over more neurons, such that there are fewer learnable weights per neuron, increases the network's expressive power. This principle pertained to LRNN, LFFN and shallow ReLU networks.

## 1.2 Related work

**Expressive Power of Neural Networks.** The expressive power of neural networks has been extensively studied. Cover (1965) established limits on the expressivity of a single perceptron,

while Cybenko (1989) and Hornik et al. (1989) demonstrated that shallow NNs serve as universal approximators. More recent work by Raghu et al. (2017); Cohen et al. (2016); Montúfar et al. (2014), highlighted the greater expressive power of deep networks compared to shallow ones. Additionally, the expressivity of specific architectures were investigated, like convolutional neural networks (CNNs) (Cohen et al., 2016), RNNs (Siegelmann and Sontag, 1995; Khrulkov et al., 2018), and graph neural networks (GNNs) (Joshi et al., 2024). Collins et al. (2017) showed that different RNN architectures, such as GRU, LSTM, and UGRNN, exhibit similar expressivity, suggesting that insights into RNN expressivity could generalize to other recurrent models. In contrast to these studies that focus on the expressivity of a fully learned model, here we will study how different allocations of a subset of parameters affect the model expressivity.

**Theory on subset learning and related techniques.** Adaptation, a technique similar to subset learning, is widely used for fine-tuning neural networks. Despite its prevalence in practice, few studies have explored the expressive power of these methods. For instance, Englert and Lazic (2022) demonstrated that neural reprogramming (Elsayed et al., 2018), a strategy that alters only the input while keeping the pretrained network unchanged, can adapt a random two-layer ReLU network to achieve near-perfect accuracy on a specific data model. Similarly, Giannou et al. (2023) examined the expressive power of fine-tuning normalization parameters, while Zeng and Lee (2024) recently analyzed the expressive power of low-rank adaptation, a concept that is reminiscent of subset learning. Furthermore, the lottery ticket hypothesis (Frankle and Carbin, 2019; Malach et al., 2020) suggests that within a neural network, subnetworks exist that are capable of matching the test accuracy of the full model.

## 2   SETTINGS AND DEFINITIONS

We consider a model $M_W$ with weights $W \in \mathbb{R}^p$:

$$y = M_W(x)$$

with $x \in \mathbb{R}^q$ and $y \in \mathbb{R}^d$. We study expressivity using a student-teacher framework (Figure 1) (Gardner and Derrida, 1989). Both models share the same architecture but have different weights, where the teacher's weights $W^*$ are fixed and known. Our goal is to find a student model that can exactly match the teacher, but with limited resources. The challenge is to match the labels produced by the teacher, which uses all $p$ weights, while the student controls only $r < p$ weights. The remaining weights are randomly drawn, but are not learned. We will show that different strategies for allocating the subset of $r$ learnable weights in the student model determine its ability to match the teacher. When considering $m$ samples, we denote the output as $Y = M_W(X)$, where $X \in \mathbb{R}^{q \times m}$ represents the $m$ inputs, and $Y \in \mathbb{R}^{d \times m}$ corresponds to the $m$ outputs produced by the model.

**Definition 2.1** *An allocation $\mathcal{A}$ for a model $M_W$ with weights $W \in \mathbb{R}^p$ is defined as the subset $\mathcal{A} \subset \{1, 2, \ldots, p\}$, which identifies the indices of the learnable weights.*

**Definition 2.2** *For an allocation $\mathcal{A}$ of size $|\mathcal{A}| = r$ and constant weights $\widehat{W} \in \mathbb{R}^{p-r}$, let $\bar{\mathcal{A}} = \{1 \leq i \leq p \mid i \notin \mathcal{A}\}$ be the complement of $\mathcal{A}$, the realization set of $\mathcal{A}$ with respect to $\widehat{W}$ is defined as:*

$$\mathcal{R}_{\mathcal{A};\widehat{W}} = \left\{ W \in \mathbb{R}^p \ s.t. \ W[\bar{\mathcal{A}}] = \widehat{W} \right\}.$$

In other words, the realization set $\mathcal{R}_{\mathcal{A};\widehat{W}}$ consists of all vectors $W$ that match the constant weights $\widehat{W}$ at the positions indexed by $\bar{\mathcal{A}}$. When the constant weights $\widehat{W}$ are clear from the context, we denote the realization set simply as $\mathcal{R}_{\mathcal{A}}$. We use this notation to define our benchmark:

**Definition 2.3** *Let $\mathcal{A}$ be an allocation of size $|\mathcal{A}| = r$. For some weights distribution $\mathcal{W}$, teacher distribution $\mathcal{T}$ and samples distribution $\mathcal{X}$, we define the **match probability** of $\mathcal{A}$ to be:*

$$MP(\mathcal{A}, m) = \Pr\left[ \exists W \in \mathcal{R}_{\mathcal{A};\widehat{W}} \ s.t. \ M_W(X) = M_{W^*}(X) \right]$$

*where $\widehat{W}$ is sampled from $\mathcal{W}$, $W^*$ is sampled from $\mathcal{T}$ and $X$ from $\mathcal{X}^m$.*

Namely, the match probability for an allocation is the probability of the student, when learning the weights allocated, to express the same $m$ labels for the same $m$ samples as the teacher. Match probability will be used as a benchmark for the expressive power of an allocation in a given model, i.e., higher $MP$ for an allocation leads to a better expressive power of the model. Since our focus is on evaluating expressivity, rather than the optimization algorithm, we will focus on the existence of such $W$, without considering the optimization process of how to find it. In addition, although we assume throughout the paper that the teacher and student share the same architecture, this assumption is not strictly required (see Appendix F). Using samples to measure expressivity is inspired by Cover (1965), where expressivity is assessed by having labeled samples and examining the probability of expressing those labels.

In this paper we investigate $MP\left(\mathcal{A}, \frac{r}{d}\right)$. As said, our goal is to use $r$ weights to make the student output the same $Y$, consisting of $dm$ independent entries. In the models we consider in this paper, if $r < dm$, a complete match is unattainable and $MP = 0$. Since we only use $MP\left(\mathcal{A}, \frac{r}{d}\right)$ we short it to $MP(\mathcal{A})$.

Throughout this paper, we will frequently encounter conditions where allocations that meet them result in $MP(\mathcal{A}) = 1$ or $MP(\mathcal{A}) = 0$. For brevity, we define 'minimal allocation' as an allocation $\mathcal{A}$ such that $MP(\mathcal{A}) = 0$, and 'maximal allocation' as an allocation $\mathcal{A}$ such that $MP(\mathcal{A}) = 1$.

**Assumptions.** In this paper, we assume that the weight distribution $\mathcal{W}$ is normal with mean 0 and that there are no correlations between the different weights. This assumption guarantees that matrices are invertible and diagonalizable. Additionally, we assume that the distributions $\mathcal{X}$ and $\mathcal{T}$ is such that any drawn $m$ samples and any $m$ output vectors of the teacher are linear independent, which means that any square matrix of inputs is invertible (see Appendix F.1 for possible relaxation of the assumptions). In few of the sections there are additional assumptions that are stated in the relevant section. Furthermore, in the linear models, we assume that the model is such that a fully-learned teacher can express any linear function. For example, in feed-forward linear network, we assume that there is no layer with size smaller than the output.

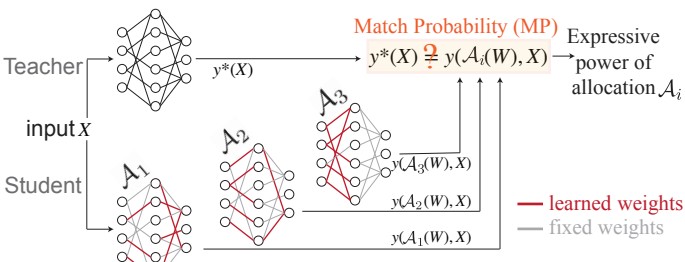

Figure 1: Schema of the student-teacher setup. The match probability (MP) estimates the expressive power of a student with an allocation $\mathcal{A}_i$ of its learnable weights by measuring its ability to match the teacher.

## 2.1 Warm up: Linear estimator model

To better understand the settings and the goal, we start with a simple example of a linear estimator, $y = Wx$, with $W \in \mathbb{R}^{d \times q}$. Considering $m$ samples we write:

$$Y = M_W(X) = WX$$

with $X \in \mathbb{R}^{q \times m}$, $Y \in \mathbb{R}^{d \times m}$. We thus ask how to allocate the $r$ learnable weights in $W$ such that the student matches the teacher successfully on all samples. For a realization of $W^*$, $X$ and $\widehat{W}$, we seek for $W \in \mathcal{R}_{\mathcal{A}; \widehat{W}}$ such that:

$$WX = W^*X$$

Denote $W_i$ and $W_i^*$ as the $i$th row of the matrices, $r_i$ the number of learnable weights in $W_i$, and $\tilde{W}_i$ as the subset of $W_i$ corresponding to these $r_i$ weights. Additionally, let $\tilde{X}$ represent the corresponding

rows of the input samples, $X$, $\widehat{W}_i$ denote the remaining (fixed) weights of $W_i$, and $\widehat{X}$ represent the remaining rows of $X$. We then have:

$$\tilde{W}_i\tilde{X} = W_i^*X - \widehat{W}_i\widehat{X} \tag{2.1}$$

where all the student and teacher weights on the r.h.s of Equation (2.1) are constants. As $\tilde{X} \in \mathbb{R}^{r_i \times m}$, if $r_i < m$ we will have more equations than variables in Equation (2.1) and there will be no solution to the set of these linear equations. In contrast, if $r_i = m$ there is always a solution:

$$\tilde{W}_i = \left(W_i^*X - \widehat{W}_i\widehat{X}\right)\tilde{X}^{-1} \tag{2.2}$$

As $r_i$ is the number of learnable weights in the $i$th row of $W$, we have $r = \sum_{i=1}^{d} r_i$. This means that an allocation must have an equal division between the rows of $W$, each row consisting of exactly $\frac{r}{d}$ learnable weights. Based on Equation (2.2) we can allocate the learnable weights to create $W \in \mathcal{R}_{\mathcal{A}}$ such that $W^*X = WX$. Therefore, in that case $MP(\mathcal{A}) = 1$, namely a maximal allocation, and otherwise $MP(\mathcal{A}) = 0$, namely minimal allocation.

Notice the dichotomy here - there is an allocation that perfectly matches the teacher, but its not robust. A slight change in the allocation, such as moving a single learnable weight from one row to another, will create an allocation that will never match the teacher (a minimal allocation). This concept will return later on.

## 3 Linear RNN model

We begin by applying our framework to study allocations of learnable weights in RNNs. We focus on RNNs both because of their relevance to modeling neural circuits, as well as the non-trivial results arising from allocating the learnable weights in their recurrent weights.

Consider a linear recurrent neural network:

$$h_t = Wh_{t-1} + Bx_t \tag{3.1}$$

$$y_t = Dh_{t-1} \tag{3.2}$$

with the hidden state $h_t \in R^n$, the initial state $h_0 = 0$, a step input $x_t \in \mathbb{R}^b$, and with $t = 1...T$. The input to the network is driven through the encoder matrix $B \in \mathbb{R}^{n \times b}$, and $W \in R^{n \times n}$ is the recurrent connections. The hidden state is read out through the decoder $D \in \mathbb{R}^{d \times n}$, such that the network's output is $y_t \in \mathbb{R}^d$. We consider $y_{T+1}$ as the output of the network, and denote $y = y_{T+1}$. The RNN is thus a function $F : \mathbb{R}^{Tb} \to \mathbb{R}^d$. We assume $n \gg b, d$. Additionally, when referring to $m$ inputs, we denote $X_t \in \mathbb{R}^{b \times m}$, where each column represents one of the $m$ inputs at step $t$ across all samples.

First, due to a key attribute of LRNN, we assume that $Tb \leq n$ and $Td \leq n$ (see Theorem B.1 for details).

Second, solving the recursion gives:

$$Y = D\sum_{t=1}^{T} W^{T-t+1}BX_t \tag{3.3}$$

This implies that for the student to match the teacher, it must solve a system of $dm = r$ polynomials (since $Y \in \mathbb{R}^{d \times m}$) with $r$ variables. Notably, the high polynomial degree arises only when allocating weights in $W$. As a result, we show that allocating the subset of learnable synapses to the encoder, decoder, or the recurrent interactions lead to very different allocation strategies.

### 3.1 Learning the decoder

We first consider allocations that learn the decoder, $D$. Denote $X' = \sum_{t=1}^{T} W^{T-t+1}BX_t$, we get that $Y = DX'$. Learning the decoder is, therefore, analogous to that of a linear estimator. The conditions for the allocation to be optimal are the same as those described in Section 2.1. We formalize that as a theorem:

**Theorem 3.1** *For any allocation $\mathcal{A}$ learning the decoder $D \in \mathbb{R}^{d \times n}$, if every row has exactly $m$ learnable weights the allocation is maximal, else it is minimal.*

## 3.2 Learning the encoder

We continue with considering allocations that learn the encoder, $B$. In this section we assume the sample distribution is element-wise i.i.d. Similar to the decoder, this scenario reduces to a system of linear equations, where the matrix is regular for some allocations and singular for others, leading to a match probability of either 1 or 0 for each allocation. However, unlike the decoder, constructing and analyzing the matrix is more involved, as $B$ is embedded within the dynamics (see Equation (3.3)) rather than appearing at the end.

**Theorem 3.2** *Any allocation $\mathcal{A}$ learning the encoder $B \in R^{n \times b}$ that follows both of the following conditions is maximal:*

1. *No row of B has more than $Tm$ learnable weights*

2. *No columns of B has more than $Td$ learnable weights*

*Every other allocation is minimal.*

Notice that as the number of learnable weights are constant, allocating too many learnable weights in a row (column) leads to insufficient learnable weights in the other rows (columns). Furthermore, as in for the decoder, each allocation exhibits a dichotomy — being either minimal or maximal — with maximal expressivity achieved when the allocation is more distributed.

## 3.3 Learning the recurrent connections

Although the proofs for allocating the learnable subset to the decoder or encoder differed, the underlying rationale was similar. In both cases, we transformed the problem into a system of linear equations, where maximality is achieved when the matrix is invertible and minimality when it is not. However, when the allocation is applied to learning the recurrent connections, as shown in Equation (3.3), the problem shifts to solving a system of polynomials of degree $T$, making it more complex than the linear cases.

We illustrate this phenomenon in Appendix B.1.1 with a simple example network, where $T = 2$. In this scenario, Equation (3.3) simplifies into two equations, one linear and one quadratic, which can be combined into a single quadratic equation, represented as $ax^2 + bx + c = 0$. It is well-known that this type of equation is solvable if and only if $b^2 \geq 4ac$. We demonstrate that for different allocations, distinct expressions for $a$, $b$, and $c$ arise, leading to varying match probabilities. Moreover, for certain allocations, we find that $a = 0$, which implies the equation is linear, and the match probability is 1. This observation raises the question of whether this phenomenon, where maximal allocations exist even though the equations are non-linear, also occurs in larger models and how to identify such allocations.

To gain insights on under which conditions allocations become maximal, we simplify the high-degree equations. The following lemma reduces the problem to a system of linear and quadratic equations:

**Lemma 3.3** *Given samples $X$ and labels $Y$ created by the teacher, the student matches them using allocation $\mathcal{A}$ if and only if the following set of equations is solvable:*

$$W g_B(F) = F \tag{3.4}$$
$$DFX = Y \tag{3.5}$$

with $W \in \mathcal{R}_{\mathcal{A}}$, $F \in \mathbb{R}^{n \times Tb}$ and $g_B$ is a linear operator that depends on $B$, formally defined in the appendix. Note that these equations are essentially Equation (3.3) (see appendix), where Equation (3.4) is just a step of the dynamics. The key insight is that by treating the dynamic steps

themselves as variables, we alter the problem's structure, making it more tractable. Introducing the $nTb$ new auxiliary variables, denoted as $F$, in addition to the variables in $W$, reduces the polynomial degree of $T$ in Equation (3.3) to a system of linear and quadratic equations in $W$ and $F$.

Generally, the equations in Equation (3.4) are quadratic, while those in Equation (3.5) are linear. However, if a full row in $W$ has no learnable weights, the $Tb$ equations corresponding to that row become linear, involving only the variables from $F$. As shown in Appendix B.1, a similar observation applies to columns. This means that if an allocation uses fewer rows or columns for its learnable weights, the resulting system will contain more linear equations and fewer quadratic ones.

One might ask: How does altering the number of rows or columns–and thereby the number of quadratic equations–affect overall solvability? First, if an allocation uses too few rows or columns, the match probability will be zero. This occurs because when there are too many equations involving only the variables from $F$, we end up with a situation where there are more variables than equations, resulting in a "waste" of variables. This idea is formalized in the following theorem:

**Theorem 3.4** *Any allocation $\mathcal{A}$ learning the recurrent connections $W \in \mathbb{R}^{n \times n}$ that follows at least one of the following conditions is minimal:*

1. *A row in $W$ has more than $Tb$ learnable weights*

2. *A column in $W$ has more than $Td$ learnable weights*

Second, when an allocation utilizes a certain number of rows or columns, the problem can simplify into a linear system, which will be solvable with probability 1. In this case, the allocation becomes maximal. This concept is formalized in the following theorem:

**Theorem 3.5** *Any allocation $\mathcal{A}$ learning the recurrent connections $W \in \mathbb{R}^{n \times n}$, that follows one of the following conditions is maximal:*

1. *Each row has $Tb$ or 0 learnable weights*

2. *Each column has $Td$ or 0 learnable weights*

Namely, allocation that uniformly distribute the learnable weights in $\frac{r}{Tb}$ rows or $\frac{r}{Td}$ columns is always maximal. Notice that this is the least number of rows and columns that allocation should use, as less rows or columns can't satisfy the conditions in Theorem 3.4. Recall that we assumed $Tb$ and $Td$ to be smaller than $n$, ensuring that the number of learnable weights does not exceed the length of any row or column.

In other cases, we still need to solve a system of quadratic equations to determine whether the student matches the teacher. The problem of determining the number of solutions for a set of polynomial equations is well studied in mathematics. For example, Smale's 17th problem (Smale, 1998) addresses the algorithmic challenges of root-finding for complex polynomials, though much less is known for polynomials with real coefficients. Recently, Subag (2024) investigated the number of solutions for a system of $n$ polynomials in $n$ variables with Gaussian coefficients and showed that the probability of finding at least one solution increases with the number of polynomials (and variables), eventually converging to 1. Therefore, we conjectured that as an allocation uses more rows—thus introducing more quadratic equations—the match probability will also increase. Moreover, as increasing the model size increases the number of quadratic equations, we also anticipate that for large networks the match probability for all allocations will approach 1.

To test this conjecture, we used numerical simulations to estimate the match probability while restricting the allocation of learning synapses to a subset of the rows. Figure 2a shows that across different network sizes the match probability increases as more rows are utilized in the allocation. Additionally, for the same percentage of rows, the match probability increases with the network size. This is consistent with the intuition that increasing the total number of polynomials (by increasing the network size) increases the probability to find a solution to the set of polynomials. Figure 2b depicts the match probability for models with constant $r/n$, which signifies the sharp transition from minimal (zero expressivity) to maximal expressivity at the $\frac{r}{Tb}$th row (see Theorems 3.4 and 3.5).

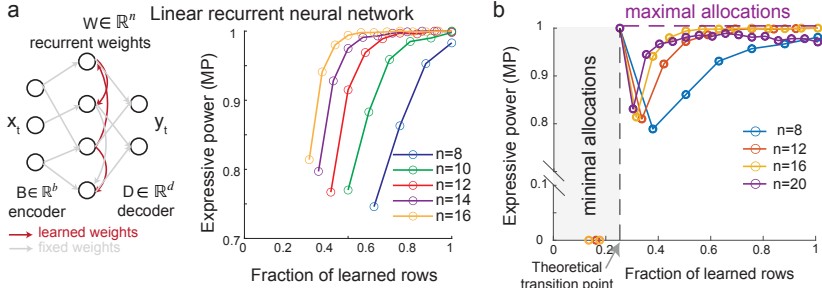

Figure 2: a. Estimation of MP for allocations in the recurrent weights of LRNN with $d = 4$ for different sizes of the hidden state, $n$. Note that MP increases with $n$. b. Same as (a), but with fixed $d/n = \frac{1}{4}$. In this case, $\frac{r}{Tb} = \frac{n}{4}$, which means that allocations using $\frac{1}{4}$ of the rows for every $n$ follow Theorem 3.5, and are thus maximal. Allocations using less rows are minimal due to Theorem 3.4. Note that MP approaches 1 as $n$ increases. All experiments ran with second order optimization methods. See Appendix E for full details.

## 4    LINEAR MULTI-LAYER FEED-FORWARD MODEL

The findings for linear RNNs provide insights into the expressive power of subset learning in linear multi-layer feedforward networks (LFFNs). In these models, the input is multiplied by several matrices. Formally:

$$Y = W_L W_{L-1} \cdots W_2 W_1 X$$

where $W_l \in \mathbb{R}^{n_l \times n_{l-1}}$, $2 \leq l \leq L - 1$, and $W_L \in \mathbb{R}^{d \times n_{L-1}}$, $W_1 \in \mathbb{R}^{n_1 \times q}$. Additionally, we require $n_l \geq q, d$ for every $1 \leq l \leq L$. This ensures that the network dimension never decreases below $q$ and $d$.

As the last layer acts as the decoder, it has indeed the same attributes as learning the decoder in the LRNN model as we saw in Section 3.1. Allocations of learnable synapses in one of the intermediate hidden layers, or the encoding layer, is reminiscent of allocations in the encoder of the LRNNs. This is formalized in the next theorem:

**Theorem 4.1** *For any allocation $\mathcal{A}$ learning an intermediate or encoder layer $W_l \in \mathbb{R}^{n_l \times n_{l-1}}$ is maximal if and only if it follows one of the following:*

1. *There is no rows that has more than $m$ learnable weights*

2. *There is no columns that has more than $d$ learnable weights*

*Otherwise, the allocation is minimal.*

Allocating learnable synapses across multiple matrices is analogous to learning the recurrent connections in LRNNs. This type of allocation results in $r$ polynomials with $r$ variables, where the degree of the polynomials corresponds to the number of learnable layers. Similarly, as we showed for allocations of the recurrent connections, certain allocations can reduce the number of polynomials involved.

Using the same framework and arguments as Lemma 3.3, one can show that the student matches the teacher iff the following equations are solvable:

$$W_1 X = F_1 \tag{4.1}$$
$$W_l F_{l-1} = F_l \quad 2 \leq l \leq L - 1 \tag{4.2}$$
$$W_L F_{L-1} = Y \tag{4.3}$$

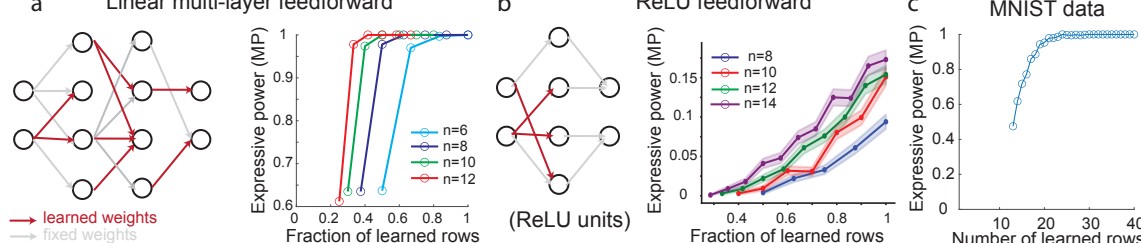

Figure 3: Estimation of MP for FFNs. MP increases when distributing the weights. a. Estimation of MP for allocations in 3-layer linear FFNs network, for different sizes of the intermediate layer denoted as $n$. b. Allocation for shallow ReLU network for different sizes of the hidden layer size. All experiments ran with second order optimization methods. c. Allocations in a shallow ReLU network with n=1000 on structured data (MNIST). The MP when the allocation uses more then 40 rows was constant 1. See Appendix E for full details.

with $W_l \in \mathbb{R}^{n_l \times n_{l-1}}$, $2 \le l \le L-1$, $W_L \in \mathbb{R}^{d \times n_{L-1}}$, $W_1 \in \mathbb{R}^{n_1 \times q}$, $F_l \in \mathbb{R}^{n_l \times m}$, $1 \le l \le L-1$. We introduce many new variables, denoted as $F$. However, if an allocation is restricted to only a small subset of rows, many of the resulting equations become linear. By solving these linear equations, we can reduce the number of polynomials to fewer than $r$.

As mentioned earlier, we expect that the probability of a set of polynomials having a solution to increase as the number of polynomials increases. Therefore, we expect that allocations restricted to a small subset of rows across different layers will have a lower match probability. This expectation is confirmed empirically, as shown in Figure 3a.

Therefore, in FFNs, similar to learning the recurrent weights in LRNNs, the multiplication of multiple matrices (due to the layers) leads to non-linear equations, which motivates distributing the learnable weights to increase the match probability.

## 5 Shallow ReLU model

We saw that generally, it is better to distribute the learnable weights throughout the network (Figures 2 and 3a). We conclude with an intuition, backed by empirical evidence, that this is also true in a shallow network of ReLU units.

Consider a 2-layer feedforward ReLU network:

$$y_i = W_2 \, \phi(W_1 x_i)$$

with $W_1 \in \mathbb{R}^{n \times q}$, $W_2 \in \mathbb{R}^{d \times n}$, and where $x_i$ and $y_i$ is the $i$'th sample and label respectively for $1 \le i \le m$. We consider allocations learning $W_1$. For the linear model, as long as some conditions hold (Theorem 4.1), all allocations are maximal. We now want to examine if the ReLU changes these results and whether it adds another incentive for distributing the learnable weights of an allocation.

For a given allocation $\mathcal{A}$, let $k \le n$ be the number of rows that has learnable weights. Assume w.l.o.g that they are the upper rows, and denote $W_1^1$ to be the first $k$ rows and $W_1^2$ to be the other $n-k$. Denote $W_2^1$ and $W_2^2$ to be the corresponding columns in $W_2$. Thus we can write:

$$W_2^1 \, \phi(W_1^1 x_i) = y_i - W_2^2 \, \phi(W_1^2 x_i)$$

Notice that the r.h.s is constant. For $\phi(x)$ being the ReLU function, this equation can be rewritten as:

$$W_2^1 P_i W_1^1 x_i = y_i - W_2^2 \, \phi(W_1^2 x_i) \tag{5.1}$$

Here, $P_i \in \mathbb{R}^{k \times k}$ is a diagonal matrix with 1s on rows where $W_1^1 x_i > 0$ and 0s where it is negative, namely $(2P_i - I)W_1^1 x_i > 0$. For a fixed $P = \{P_i\}_{i=1}^m$ we get a linear equation in the $r$ learning

weights of $W_1^1$. Thus, for the allocation to match, two conditions must hold: 1) the linear equation must be solvable, and 2) the solution of $W_1^1$ must satisfy $(2P_i - I)W_1^1 x_i > 0$ for every $x_i$.

The first observation is that if the allocation does not satisfy the conditions in Theorem 4.1, the linear equations will be unsolvable. Another key point is that we need to "guess" $P$, which could, in theory, take one of $2^{km}$ possible configurations, solve for $W_1^1$, and verify if it satisfies the inequalities. However, the actual number of feasible configurations is lower—Lemmas D.3 and D.4 demonstrate that some configurations are inherently invalid due to the linear equations being unsolvable. Importantly, the fraction of feasible configurations increases as the allocation becomes more distributed—Lemma D.3 shows that the number of valid configurations for $P$ decreases when smaller values of $k$ are used, and Lemma D.4 reveals that allocations with rows containing many learnable weights similarly face a reduction in the number of possible configurations for $P$.

This suggests that allocations using more rows are more likely to increase the match probability. As shown in Figure 3b, the empirical trend aligns with our intuition. These results suggest that the ReLU non-linearity introduces an added benefit for allocations that distribute their learnable weights more broadly across the network.

## 6 Discussion

In this paper, we explored different strategies for learning under resource constraints, focusing on how to allocate a subset of learnable model weights from a standpoint of expressivity. Our main focus was on linear RNNs and FFNs. Linear models have been fundamental across fields such as linear circuits and control theory, despite being oversimplifications of real-world systems. Their strength lies in simplifying complex ideas, providing a necessary foundation for understanding more intricate nonlinear behaviors (Saxe et al., 2014). Although the linearity of these models, we showed that the problem of how to allocate the resources when examining subset learning becomes highly non-linear, with linearity emerging only in specific cases (Theorems 3.4 and 3.5).

We conjectured that the number of non-linear equations arising from subset learning in linear models improves model expressivity, and that distributing the learnable weights increases the number of non-linear equations, thus generally improves the expressive power of the model. This conjecture was backed-up with numerical simulations (Figures 2 and 3). Interestingly, and perhaps unexpectedly, our results suggests that in large models, allocations can consistently achieve maximum expressivity.

While our work provides theoretical insights into the impact of allocation strategies on expressivity, several limitations should be acknowledged. First, our paper should be viewed as a first step in exploring how allocation strategies affect the network performance. As such, our analysis primarily focuses on expressivity, which does not encompass other important aspects of network performance, such as generalization or optimization, and these should be explored in future work. Additionally, our theoretical framework relies on assumptions like i.i.d. inputs and linear independence of certain matrices, which may not fully capture the complexities of practical neural networks. Another limitation is that many results were derived for simplified architectures, necessary to develop insights beyond numerical investigations. We took a step toward generalizing the results by extending our framework to study expressivity in a non-linear shallow ReLU model, where we found that non-linearity itself motivates distributed allocations. Future work should extend these results numerically and theoretically to more complex architectures and structured datasets.

Despite these limitations, our paper presents a general framework for studying expressivity in neural networks. Match probability (MP) is a versatile and applicable measurement that captures the reduction in expressivity due to specific allocation strategies. While we use a student-teacher framework for clarity, our analysis does not rely on the teacher model itself, and MP can be naturally defined for general data distribution (see Appendix F). In fact, we tested our approach on structured, real-world data by estimating the MP of a shallow ReLU network on the MNIST dataset (Figure 3c). These results suggest that our insights regarding distributing the learnable weights throughout the network may pertain to practical applications.

## 7 Acknowledgements

We thank Oren Yakir, Lior Bary-Soroker and Roi Livni for their input and constructive feedback. We thank Tomer Koren for his input, constructive feedback and carefully reading the manuscript. This work was supported by research grants from the Israel Science Foundation (2097/24, 2994/24)

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

# Appendix

## A  LINEAR ALGEBRA PRELIMINARIES: KRONECKER PRODUCT AND VECTORIZATION

We use $I_n$ as the identity matrix of size $n \times n$. When the size is clear from context we write $I$.

**Definition A.1**  *The Kronecker product of two matrices $A$ and $B$ is denoted by $A \otimes B$ and is defined as follows:*

*Let $A$ be an $m \times n$ matrix and $B$ be a $p \times q$ matrix:*

$$A = \begin{pmatrix} a_{11} & a_{12} & \cdots & a_{1n} \\ a_{21} & a_{22} & \cdots & a_{2n} \\ \vdots & \vdots & \ddots & \vdots \\ a_{m1} & a_{m2} & \cdots & a_{mn} \end{pmatrix}_{m \times n} , \quad B = \begin{pmatrix} b_{11} & b_{12} & \cdots & b_{1q} \\ b_{21} & b_{22} & \cdots & b_{2q} \\ \vdots & \vdots & \ddots & \vdots \\ b_{p1} & b_{p2} & \cdots & b_{pq} \end{pmatrix}_{p \times q}$$

*The Kronecker product $A \otimes B$ is an $(mp) \times (nq)$ matrix defined by:*

$$A \otimes B = \begin{pmatrix} a_{11}B & a_{12}B & \cdots & a_{1n}B \\ a_{21}B & a_{22}B & \cdots & a_{2n}B \\ \vdots & \vdots & \ddots & \vdots \\ a_{m1}B & a_{m2}B & \cdots & a_{mn}B \end{pmatrix}$$

**Definition A.2** *Let $A$ be an $m \times n$ matrix. The vectorization of $A$, denoted by $vec(A)$, is the $mn \times 1$ column vector obtained by stacking the columns of $A$ on top of one another. For example:*

$$A = \begin{pmatrix} a_{11} & a_{12} & \cdots & a_{1n} \\ a_{21} & a_{22} & \cdots & a_{2n} \\ \vdots & \vdots & \ddots & \vdots \\ a_{m1} & a_{m2} & \cdots & a_{mn} \end{pmatrix} \qquad vec(A) = \begin{pmatrix} a_{11} \\ a_{21} \\ \vdots \\ a_{m1} \\ a_{12} \\ a_{22} \\ \vdots \\ a_{m2} \\ \vdots \\ a_{1n} \\ a_{2n} \\ \vdots \\ a_{mn} \end{pmatrix}$$

We will use two well known attributes of Kronecker product (see Horn and Johnson (1991)):

**Lemma A.3** *For every $A \in \mathbb{R}^{d_1 \times d_2}$, $B \in \mathbb{R}^{d_2 \times d_3}$, $C \in \mathbb{R}^{d_3 \times d_4}$:*

$$vec(ABC) = (C^T \otimes A)vec(B)$$

**Lemma A.4** *For every $A \in \mathbb{R}^{d_1 \times d_2}$, $B \in \mathbb{R}^{d_3 \times d_4}$:*

$$rank(A \otimes B) = rank(A)rank(B)$$

## B  LINEAR RNN

We start with a key feature of LRNNs: although the total number of weights is $p = (d + b + n)n$, only $p = (d + b)n$ of these are necessary to represent the network's output. This result is formalized in the following theorem:

**Theorem B.1** *For every $T$, the output of the network can be expressed using $n(d + b)$ variables.*

**Proof:** From Equation (3.3) we get that if the value of $DW^{T+1-t}B$ for two networks is the same for all $T$, their output will always be equal.

The measure of real matrices that are not diagonalizable over the complex equals 0, so the probability for a random matrix with a continuous probability distribution to be non-diagonalizable vanishes. Since $W$ is such matrix, we can safely assume it diagonizable. Let $U\lambda U^{-1}$ be the spectral decomposition of $W$.

For every diagonal matrix $S$, we can write:

$$W = US\lambda S^{-1}U^{-1} = (US)\lambda(US)^{-1}$$

Now fix $S$ such that $D_1 US = \overrightarrow{1}$, where $\overrightarrow{1}$ is a vector of all 1's and $D_1$ is the first row of $D$. Denote $V = US$, we have $W = V\lambda V^{-1}$ such that $D_1 V = \overrightarrow{1}$. In other words, we can always pick the eigenvectors $V$ of $W$ to be such that $D_1 V = \overrightarrow{1}$.

Notice that $DW^{T+1-t}B = DV(\lambda^{T+1-t})V^{-1}B$. Thus, two networks that will have the same $DV$, $V^{-1}B$ and $\lambda$ will always output the same system. There are $bn$ variables in $V^{-1}B$, and $n$ variables in $\lambda$. $DV$ has $dn$ entries but we know that the first row is $\overrightarrow{1}$ so it has only $dn - n$. Thus, these $n(d + b)$ variables express the output of the network for every $T$. ∎

From solving the recursion it becomes clear that LRNN is a linear transformation, receiving $Tb$ inputs and producing $d$ outputs, effectively acting as a $Tb \times d$ matrix. From the proof, we observed that this function (expressed as $DW^{T+1-t}B$ for each $1 \le t \le T$) can be represented using $n(d+b)$ variables. This implies that the the function that the network represents becomes degenerate if $Tbd > n(d+b)$. To prevent degeneration, we require $T \le \frac{n(d+b)}{db}$. Given the assumption that $n \gg b, d$, this simplifies to approximately $Tb \le n$ and $Td \le n$.

### B.1 LEARNING THE RECURRENT CONNECTIONS

In this section, we provide proofs for Theorem 3.4 and Theorem 3.5, and demonstrate their application through a small network example.

**Definition B.2 (Shift Operator)** *Given $n_1, n_2, n_3 \in \mathbb{N}$, $B \in \mathbb{R}^{n_1 \times n_2}$ and $F \in \mathbb{R}^{n_1 \times n_2 n_3}$. Denote $F = (\ F_1 \mid F_2 \mid \cdots \mid F_{m-1} \mid F_m\ )$ such that for every $1 \le i \le n_3$, $F_i \in \mathbb{R}^{n_1 \times n_2}$. Then:*

$$g_B(F) = (\ F_2 \mid F_3 \mid \cdots \mid F_{m-1} \mid F_m \mid B\ )$$

**Lemma B.3** *Given samples $X \in \mathbb{R}^{Tb \times m}$ and labels $Y \in \mathbb{R}^{d \times m}$ created by the teacher, the student can express them using allocation $\mathcal{A}$ if and only if the following set of equations is solvable:*

$$W g_B(F) = F \tag{B.1}$$
$$DFX = Y \tag{B.2}$$
$$W \in \mathcal{R}_{\mathcal{A}},\ F \in \mathbb{R}^{n \times Tb}$$

**Proof:** We will start by showing that if the system is solvable, then the student can express the given labels. Assume the equations are solvable using $W^* \in \mathcal{R}_{\mathcal{A}}$ and $F^* \in \mathbb{R}^{n \times Tb}$, $F_t^* \in \mathbb{R}^{n \times b}$. Notice that $W^*$ and $F^*$ are assumed to be solutions to the above equations and are unrelated to the teacher.

From the definition of the operator $g_B$, Equation (B.1) becomes:

$$W^* F_t^* = F_{t-1}^* \quad 2 \le t \le T$$
$$W^* B = F_T^*$$

Solving the recursion we get $F_t^* = (W^*)^{T-t+1} B$.

Since $FX = \sum_{t=1}^T F_t X_t$, Equation (B.2) is equivalent to:

$$Y = D \sum_{t=1}^T F_t X_t = D \sum_{t=1}^T (W^*)^{T-t+1} B X_t$$

Since $W^*$ is a valid realization of $\mathcal{A}$ and it outputs the correct labels, the student can match the teacher.

Now, on the other hand, assume that the student can express $Y$. We choose $W$ to be the recurrent connections of that student, which implies $Y = D \sum_{t=1}^T W^{T-t+1} B X_t$. By selecting $F_t = W^{T-t+1} B$, it is easy to verify that the equations hold. ∎

**Lemma B.4** *For any allocation $\mathcal{A}$, if $|\mathcal{A}| < dm$ then $MP(\mathcal{A}) = 0$.*

**Proof:** From Equation (3.3), the student can match with allocation $\mathcal{A}$ if a set of $dm$ polynomials is solvable with $|\mathcal{A}|$ variables. Thus, if the student can match with $\mathcal{A}$, a set of $dm$ polynomials is solvable with less than $dm$ variables. This is generally solvable w.p 0, for example from Theorem 6.8 of Azas and Wschebor (2009) this is true for polynomials with Gaussian coefficients. Therefore, the match probability of $\mathcal{A}$ is 0. ∎

**Theorem B.5 (First part of Theorem 3.4)** *Given an allocation $\mathcal{A}$, if there is a row with more than $Tb$ learnable weights the allocation is minimal.*

**Proof:** We will prove that if the student can match with an allocation with more than $Tb$ learnable weights in some row, it can match with an allocation with less than $dm$ weights, which means from Lemma B.4 that the match probability is 0.

Given an allocation $\mathcal{A}$ with $|\mathcal{A}| = r = dm$ that has $r_i > Tb$ learnable weights in the $i$th row and solves Lemma B.3. Notice that the rows of $W_i$ in Lemma B.3 participate in exactly $Tb$ equations in Equation (B.1) - the $i$th row only need to suffice:

$$W_i g_B(F) = F_i$$

Where $F_i$ is the $i$th row of $F$.

Denote the solution as $F = F^*$ and $W = W^*$. Consider another allocation, $\mathcal{A}_2$, in which we replace $r_i - Tb > 0$ of the learnable weights of $\mathcal{A}$ in the $i$'th row with constants, which means that $|A_2| < dm$. Define $\tilde{W}_i$ as the learnable part of $W_i$, $\tilde{g}_B(F^*)$ as the corresponding columns in $g_B(F^*)$, $\widehat{W}_i$ as the remaining part of $W_i$, and $\widehat{g}_B(F^*)$ as the remaining part of $g_B(F^*)$. Then:

$$\tilde{W}_i \tilde{g}_B(F^*) + \widehat{W}_i \widehat{g}_B(F^*) = F_i^*$$
$$\tilde{W}_i \tilde{g}_B(F^*) = F_i^* - \widehat{W}_i \widehat{g}_B(F^*)$$
$$\tilde{W}_i = \left(F_i^* - \widehat{W}_i \widehat{g}_B(F^*)\right)(\tilde{g}_B(F^*))^{-1}$$

Which means that there is a solution to Lemma B.3 with $\mathcal{A}_2$. In other words, every time Lemma B.3 is solvable with $\mathcal{A}$, it is also solvable with $\mathcal{A}_2$. Recall that $|\mathcal{A}_2| < dm$, therefore, from Lemma B.4, the match probability of $\mathcal{A}_2$ is 0, which means that the match probability of $\mathcal{A}$ is also 0.

∎

**Theorem B.6 (First part of Theorem 3.5)** *Given an allocation $\mathcal{A}$, if every row that has learnable weights has exactly $Tb$ learnable weights in it, the allocation is maximal.*

**Proof:** The rank of the linear equation in Equation (B.2) is $dm$, which follows immediately from Lemma A.3 and Lemma A.4 (recall that $m \leq Tb$ and $d \leq n$).

With $r$ learnable weights in the model, the allocation consists of exactly $\frac{r}{Tb} = \frac{dm}{Tb}$ rows, each containing $Tb$ learnable weights, while the remaining rows are fully constant. Without loss of generality, assume that these constant rows are positioned at the bottom of $W$. We can express the first equation of Lemma B.3 as:

$$W_{TOP} \cdot g_B(F) = F_{TOP} \tag{B.3}$$
$$W_{BOT} \cdot g_B(F) = F_{BOT} \tag{B.4}$$

Where the $TOP$ rows of $W$ are the rows with learnable weights and the $BOT$ are the ones without.

Using Lemma A.3, Equation (B.2) becomes:

$$(X \otimes D)vec(F) = Y \tag{B.5}$$

And Equation (B.4) becomes:

$$(I_{Tb} \otimes W_{BOT})vec(g_B(F)) = F_{BOT} \tag{B.6}$$

From Lemma A.4 the rank of Equation (B.5) is $dm$ and the rank of Equation (B.6) is $Tb\left(n - \frac{dm}{Tb}\right) = nTb - dm$. These equations involve only the $nTb$ variables of $F$, without including any learnable weights from $W$.

This implies that we have $nTb$ linear independent equations with $nTb$ variables, leading to a unique solution for $F$. Once we obtain a solution for $F$, we can substitute it into Equation (B.3), transforming it into a linear equation as well—each row of $W_{TOP}$ contains $Tb$ linear equations corresponding to the size of $F$. Since every row of $W_{TOP}$ contains precisely $Tb$ variables, the equation is solvable. ∎

The proof of the second condition in both Theorem 3.4 and Theorem 3.5 has the same structure as the first condition. It uses the shift operator, while introducing the auxiliary variables $F$, but now with the decoder matrix, $D$, instead of the encoder matrix, $B$.

**Lemma B.7** *Given samples $X \in \mathbb{R}^{Tb \times m}$ and labels $Y \in \mathbb{R}^{d \times m}$ created by the teacher, the student can express them using allocation $\mathcal{A}$ if and only if the following set of equations is solvable:*

$$g_D(F)W = F \tag{B.7}$$
$$F(I_T \otimes B)X = Y \tag{B.8}$$
$$W \in \mathcal{R}_{\mathcal{A}}, \, F \in \mathbb{R}^{d \times nT}$$

**Proof:** We will start by showing that if the system is solvable, then the student can express the given labels. Assume the equations are solvable using $W^* \in \mathcal{R}_{\mathcal{A}}$ and $F^* \in \mathbb{R}^{d \times nT}$, $F_t^* \in \mathbb{R}^{d \times n}$. Notice that $W^*$ and $F^*$ are assumed to be solutions to the above equations and are unrelated to the teacher.

From the definition of the operator $g_D$, Equation (B.7) becomes:

$$F_t^* W^* = F_{t-1}^* \quad 2 \le t \le T$$
$$DW^* = F_T^*$$

Solving the recursion we get $F_t^* = D(W^*)^{T-t+1}$.

Notice that:

$$(I_T \otimes B)X = \begin{pmatrix} BX_1 \\ BX_2 \\ \vdots \\ BX_{m-1} \\ BX_m \end{pmatrix}$$

$$Y = F^*(B \otimes I_T)X = \sum_{t=1}^{T} D(W^*)^{T-t+1}BX_t$$

Since $W^*$ is a valid realization of $\mathcal{A}$ and it outputs the correct labels, the student can match the teacher.

Now, on the other hand, assume that the student can express $Y$. We choose $W$ to be the recurrent connections of that student, which implies $Y = D\sum_{t=1}^{T} W^{T-t+1}BX_t$. By selecting $F_t = DW^{T-t+1}$, it is easy to verify that the equations hold. ∎

**Theorem B.8 (Second part of Theorem 3.4)** *Given an allocation $\mathcal{A}$, if there is a column with more than $Td$ learnable weights the allocation is minimal.*

**Proof:** We will prove that if the student can match with an allocation with more than $Td$ learnable weights in some column, it can match with an allocation with less than $dm$ weights, which means from Lemma B.4 that the match probability is 0.

Given an allocation $\mathcal{A}$ with $|\mathcal{A}| = r = dm$ that has $r_i > Td$ learnable weights in the $i$th column and solves Lemma B.7. Notice that the column vector $W_i$ in Lemma B.7 participate in exactly $Td$ equations in Equation (B.7) - the $i$th row only need to suffice:

$$g_D(F)W_i = F_i$$

Where $F_i$ is the $i$th row of $F$.

Denote the solution as $F = F^*$ and $W = W^*$. Consider another allocation, $\mathcal{A}_2$, in which we replace $r_i - Td > 0$ of the learnable weights of $\mathcal{A}$ in the $i$'th column with constants, which means that $|\mathcal{A}_2| < dm$. Define $\tilde{W}_i$ as the learnable part of $W_i$, $\tilde{g_D}(F^*)$ as the corresponding rows in $g_D(F^*)$, $\widehat{W}_i$ as the remaining part of $W_i$, and $\widehat{g_D}(F^*)$ as the remaining part of $g_D(F^*)$. Then:

$$\tilde{g_D}(F^*)\tilde{W}_i + \widehat{g_D}(F^*)\widehat{W}_i = F_i^*$$
$$\tilde{g_D}(F^*)\tilde{W}_i = F_i^* - \widehat{g_D}(F^*)\widehat{W}_i$$
$$\tilde{W}_i = (\tilde{g_D}(F^*))^{-1}\left(F_i^* - \widehat{g_D}(F^*)\widehat{W}_i\right)$$

Which means that there is a solution to Lemma B.7 with $\mathcal{A}_2$. In other words, every time Lemma B.7 is solvable with $\mathcal{A}$, it is also solvable with $\mathcal{A}_2$. Recall that $|\mathcal{A}_2| < dm$, therefore, from Lemma B.4, the match probability of $\mathcal{A}_2$ is 0, which means that the match probability of $\mathcal{A}$ is also 0.

$\blacksquare$

**Theorem B.9 (Second part of Theorem 3.5)** *Given an allocation $\mathcal{A}$, if every column that has learnable weights has exactly $Td$ learnable weights in it, the allocation is maximal.*

**Proof:** With $r$ learnable weights in the model, the allocation consists of exactly $\frac{r}{Td} = \frac{dm}{Td} = \frac{m}{T}$ columns, each containing $Td$ learnable weights, while the remaining columns are fully constant. Without loss of generality, assume that these constant columns are positioned at the right side of $W$. We can express the first equation of Lemma B.7 as:

$$g_D(F) \cdot W_{LEFT} = F_{LEFT} \tag{B.9}$$

$$g_D(F) \cdot W_{RIGHT} = F_{RIGHT} \tag{B.10}$$

Where the $LEFT$ columns of $W$ are the columns with learnable weights and the $RIGHT$ are the ones without.

The matrix $W_{\text{RIGHT}}$ is of size $\left(n - \frac{m}{T}\right) \times n$, meaning that Equation (B.10) represents a linear equation (in $F$) with a rank of $n - \frac{m}{T}$ for each row of $g_D(F)$. Given that there are $Td$ such rows, we obtain a total of $nTd - dm$ linear independent equations (in the same way obtained in the proof of Theorem B.6). When we include Equation (B.8), which is a linear equation of rank $dm$, the cumulative total becomes $nTd$ linear equations. These equations involve only the $nTd$ variables of $F$, without including any learnable weights from $W$.

This implies that we have $nTd$ linear equations with $nTd$ variables, leading to a unique solution for $F$. Once we obtain a solution for $F$, we can substitute it into Equation (B.9), transforming it into a linear equation as well—each column of $W_{\text{LEFT}}$ contains $Td$ linear equations corresponding to the size of $F$. Since every column of $W_{\text{LEFT}}$ contains precisely $Td$ variables, the equation is solvable. $\blacksquare$

### B.1.1 Example small model

Here, we present a small model as an example that illustrates the different phenomena arising in subset learning of the recurrent connections of LRNN. For this example, we use the following weights: $b = 1$, $d = 1$, $n = 2$, $T = 2$, and $m = 2$.

Since $m = Tb$, this implies that the matrix $X$ is a square matrix, making it invertible. From Equation (3.3), we know that the LRNN model can eventually be reduced to a linear function. Let the matrix representing the linear function of the teacher be denoted by $A^*$ and for the student by $A$. If the student successfully matches the teacher, this implies:

$$AX = A^*X$$

Given that $X$ is invertible, this leads to $A = A^*$. Thus, we can disregard $X$, as the student will match the teacher if and only if $A = A^*$. The matrix $A^* \in \mathbb{R}^{d \times Tb}$ is a $1 \times 2$ matrix, meaning that $A^*$ is a vector with two entries, denoted as $\{A_1^*, A_2^*\}$. Therefore, the student matches the teacher if:

$$DWB = A_1^* \tag{B.11}$$

$$DW^2B = A_2^* \tag{B.12}$$

For each possible allocation, solving Equation (B.11) eliminates one variable, and Equation (B.12) becomes a quadratic equation in one variable. We denote this equation as $ax^2 + bx + c = 0$, where $a$, $b$, and $c$ are random variables that are derived from expressions involving all other random variables (i.e., $D$, $B$, $A^*$, and the constant part of $W$). The equation is solvable if and only if $b^2 \geq 4ac$. Thus, the match probability is precisely the probability that $b^2 \geq 4ac$.

There are $\binom{4}{2} = 6$ possible allocations for the $r = 2$ weights in $W$. Interestingly, in four of these allocations, when manually calculating the expression for $a$, all coefficients reduce to yield $a = 0$. This

simplifies the equation to a linear one, making it always solvable. These allocations are, therefore, maximal. As expected, these are the four allocations that satisfy the conditions of Theorem 3.5.

The two allocations that do not meet the conditions of Theorem 3.5 occur when the diagonal or off-diagonal entries are allocated. Specifically, if:

$$W = \begin{pmatrix} W_1 & W_2 \\ W_3 & W_4 \end{pmatrix}$$

The two suboptimal allocations are $\{W_1, W_4\}$ (the diagonal allocation) and $\{W_2, W_3\}$ (the off-diagonal allocation).

We evaluated the match probability for these allocations, drawing all weights from a normal distribution with variance as described in Appendix E.3. The match probability, $MP(\mathcal{A})$, was 0.83 for the off-diagonal allocation and 0.74 for the diagonal allocation. The code for this analysis is included in the attached zip file under the scripts `example_model_diagonal.py` and `example_model_off_diagonal.py`.

This example reveals two key phenomena. First, unlike allocations involving the decoder and encoder, when learning the recurrent connections, there exist sub-optimal, non-minimal allocations due to the non-linearity of the equations. Second, even with non-linear equations, maximal allocations still exist.

### B.2  LEARNING THE ENCODER

In this section, we assume that $X$ is i.i.d. and not drawn from some underlying sample distribution $\mathcal{X}$. This assumption is unnecessary in other sections because when $B$ is constant and uncorrelated with $X$, we can treat the input as $BX_t$, effectively eliminating any row-wise correlations (Corollary B.11). However, since $B$ is learned in this context, the correlations within $X$ will significantly impact the maximality conditions of $\mathcal{A}$. Therefore, to avoid these complications, we assume no correlation in $X$ for this section.

We start with a few useful lemmas before proving the main theorem in Theorem B.14. A simpler but conceptually similar proof can be found in Theorem C.1. Therefore, the proof of Theorem C.1 can be considered as a proof sketch for this theorem.

**Lemma B.10**  *Let $v_1, v_2, v_3 \in \mathbb{R}^n$ be uncorrelated vectors with mean 0. Then, $\langle v_1, v_2 \rangle$ is uncorrelated to $\langle v_1, v_3 \rangle$*

**Proof:**

$$\mathbb{E}\left[\sum_{ij}^n v_i^1 v_i^2 v_j^1 v_j^3\right] = \mathbb{E}\left[\sum_i^n (v_i^1)^2 v_i^2 v_i^3\right] = \sum_i^n E\left[(v_i^1)^2 v_i^2 v_i^3\right] = \sum_i^n E\left[(v_i^1)^2\right] E\left[v_i^2\right] E\left[v_i^3\right] = 0$$

∎

**Corollary B.11**  *Let $A$ be matrices such that there are no correlations between its rows. Then there are no correlation between the rows of $AB$.*

**Lemma B.12**  *Let $V = \{V_i \in \mathbb{R}^{m \times k}\}$ and $U = \{U_i \in \mathbb{R}^{k \times d}\}$ be two sets of matrices such that any two entries of the matrices are uncorrelated. Let $\tilde{V} = \{\tilde{V}_i \in V\}_{i=1}^{dm}$ be a sequence of size $dm$ containing matrices from $V$, and similarly, $\tilde{U} = \{\tilde{U}_i \in U\}_{i=1}^{dm}$ be a sequence containing matrices from $U$. Denote $S = \{\tilde{V}_i \tilde{U}_i\}_{i=1}^{dm}$ as a sequence of matrices in $\mathbb{R}^{m \times d}$ such that $|S| = dm$ (i.e., there is no $i, j$ such that $\tilde{V}_i \tilde{U}_i = \tilde{V}_j \tilde{U}_j$).*

*$S$ is almost surely a basis if every entry of $V$ appears in $\tilde{V}$ no more than $kd$ times and every entry of $U$ appears in $\tilde{U}$ no more than $md$ times. Otherwise, it is not a basis.*

**Proof:**

**First direction** Assume there exists $v \in V$ such that $v$ appears in $\tilde{V}$ $kd + 1$ times. Without loss of generality, assume it is the first $kd + 1$ entries of $\tilde{V}$ (i.e., $v = \tilde{V}_1 = \tilde{V}_2 = \cdots = \tilde{V}_{kd+1}$). We will show that $S$ is not a basis.

Let $\alpha \neq 0 \in \mathbb{R}^{kd+1}$ be a vector such that:

$$\sum_{i=1}^{kd+1} \alpha_i \tilde{U}_i = 0$$

Such a vector $\alpha$ exists because $U$ consists of $\mathbb{R}^{k \times d}$ matrices, so any $kd + 1$ matrices are linearly dependent. Thus:

$$\sum_{i=1}^{kd+1} \alpha_i S_i = \sum_{i=1}^{kd+1} \alpha_i \tilde{V}_i \tilde{U}_i = \tilde{V}_i \sum_{i=1}^{kd+1} \alpha_i \tilde{U}_i = 0$$

This implies that there are linearly dependent vectors in $S$, meaning $S$ is not a basis. The proof is exactly the same for the case where there exists $u \in U$ such that $u$ appears in $\tilde{U}$ $md + 1$ times.

**Second direction** Assume that every $v \in V$ appears in $\tilde{V}$ no more than $dk$ times, and every $u \in U$ appears in $\tilde{U}$ no more than $dm$ times. We aim to show that, almost surely, $S$ is a basis, meaning the only $\alpha \in \mathbb{R}^{dm}$ satisfying:

$$\sum_{i=1}^{dm} \alpha_i S_i = 0 \tag{B.13}$$

is $\alpha = 0$.

We will prove this by induction on the number of different entries from $V$ and $U$ used in $S$.

**Base case:** The minimal number of different $V$'s is $\frac{m}{k}$. We will show that $S$ forms a basis in this case; the same proof applies for a minimal number of $\tilde{U}$'s. Assume, without loss of generality, that the first $dk$ entries of $\tilde{V}$ are the same, followed by the next $dk$, and so on. Then:

$$\sum_{i=1}^{dm} \alpha_i S_i = \sum_{i=1}^{dm} \alpha_i \tilde{V}_i \tilde{U}_i = \sum_{i=1}^{\frac{m}{k}} V_i \sum_{j=1}^{kd} \alpha_{ikd+j} \tilde{U}_{ikd+j}$$

Since for every $i$, the set $\{\tilde{U}_{ikd+j}\}_{j=1}^{kd}$ is linearly independent, we have $\alpha_{ikd+j} = 0$ for every $i, j$, implying that $\alpha = 0$.

**Inductive step:** Assume the statement holds for $n$ different $v \in V$ and $u \in U$ in $S$, and prove it for $n + 1$. Assume by contradiction that $\alpha \neq 0$. Let $v \in V$ such that there exists $1 \leq i \leq dm$ where $\tilde{V}_i = v$ and $\alpha_i \neq 0$, meaning $v$ is part of the linear combination in (B.13).

Denote $U_v = \{u \in U \mid \exists s \in S \text{ such that } s = uv\}$, i.e., the subset of $U$ corresponding to entries paired with $v$. Define $S_v = \{uv \mid \forall u \in U_v\}$, the subset of $S$ containing $v$. The probability that only elements in $S_v$ have non-zero coefficients in (B.13) is 0 because $U_v$ is linearly independent (since $|U_v| \leq kd$ and the vectors in $U_v$ are uncorrelated).

Denote $L_{U_v}$ as the set of all linear subspaces in $\mathbb{R}^{m \times d}$ formed by every subset of the entries of $U_v$. If $S \setminus S_v$ spans one of the linear spaces in $L_{U_v}$, this contradicts the induction assumption, as this would imply a linear dependence with $n$ different matrices in $S$.

If $S \setminus S_v$ does not span $L_{U_v}$, the set of linear subspaces of dimension $|S_v|$ in $L_{U_v}$ that contain vectors in $\text{span}(S \setminus S_v)$ is a null set. Since $\text{span}(S_v)$ is such a subspace, and is uncorrelated with $\text{span}(S \setminus S_v)$, the probability of having some $v' \in \text{span}(S_v)$ such that $v' \in \text{span}(S \setminus S_v)$ is 0. Therefore, the probability of this scenario occurring is indeed 0. ∎

**Lemma B.13** *For any $a, b \leq T$ and $1 \leq i, j, i', j' \leq n$, the correlation between $W_{i,j}^a$ and $W_{i',j'}^b$ is 0 or aiming to 0 as $n$ aims to $\infty$.*

**Proof:** Isserlis' theorem states that for any $n$ gaussian random variables $X_1, X_2, \ldots X_n$:

$$\mathbb{E}[X_1 X_2 \cdots X_n] = \sum_{p \in \mathcal{P}_\backslash^\in} \prod_{i,j \in p} \mathbb{E}[X_i X_j]$$

Where the sum is over all the pairing of $\{1, 2, \ldots n\}$, i.e all distinct way of partitioning $\{1, 2, \ldots n\}$, into pairs $\{i, j\}$, and the product is over the pairs contained in $p$.

Now prove the lemma.

$$W_{i,j}^a = \sum_{p_1, p_2, \ldots p_{a-1}} W_{i p_1} W_{p_1 p_2} \ldots W_{p_{a-1} j}$$

$$W_{i',j'}^b = \sum_{q_1, q_2, \ldots q_{b-1}} W_{i' q_1} W_{q_1 q_2} \ldots W_{q_{b-1} j'}$$

$$\mathbb{E}\big[\langle W_{i,j}^a, W_{i',j'}^b \rangle\big] = \sum_{p_1, p_2, \ldots p_{a-1}} \sum_{q_1, q_2, \ldots q_{b-1}} \mathbb{E}\big[W_{i p_1} W_{p_1 p_2} \ldots W_{p_{a-1} j} W_{i' q_1} W_{q_1 q_2} \ldots W_{q_{b-1} j'}\big]$$

Denote $c = a + b$. Fix some $p_1, p_2, \ldots p_{a-1}$ and $q_1, q_2, \ldots q_{b-1}$. From Isserlis' theorem, in order to calculate $\mathbb{E}\big[W_{i p_1} W_{p_1 p_2} \ldots W_{p_{a-1} j} W_{i' q_1} W_{q_1 q_2} \ldots W_{q_{b-1} j'}\big]$, we need to count the number of possible pairings. Since there are $c$ variables in the product, the number of pairs is:

$$\frac{c!}{2^{\frac{c}{2}} \frac{c}{2}!}$$

This can be obtained using a simple combinatorical calculation.

Now fix a specific pairing of the $c$ variables. Since the entries of $W$ are uncorrealted, the expectation of the sum of different entries is 0. Thus, only if all the pairs in the pairing has two identical variables add value to the sum. Assume the pair of $W_{i p_1}$ is $W_{p_k p_l}$, this means that $i = p_k$. Since $p_k$ is also in another pair, it will enforce another variable to be equal to $i$. This will end when the pair will be $W_{p_{a-1} j}$ or $W_{q_{b-1} j'}$. In the first case it requires $i = j$ and in the latter $i = j'$. Another such chain will start from the pair of $W_{i' q_1}$, and again it requires $i' = j$ or $i' = j'$.

Therefore, for any pairing there is no more then single entry in the sum $\sum_{p_1, p_2, \ldots p_{a-1}} \sum_{q_1, q_2, \ldots q_{b-1}}$ that makes this pairing non-zero in expectation. Since the variance of the entries of $W$ is $\frac{1}{n}$, the variance of the product of the pairs is $n^{-c/2}$, which means:

$$\mathbb{E}\big[\langle W_{i,j}^a, W_{i',j'}^b \rangle\big] \leq \frac{c!}{2^{\frac{c}{2}} \frac{c}{2}! n^{\frac{c}{2}}}$$

From Stirling inequality:

$$\mathbb{E}\big[\langle W_{i,j}^a, W_{i',j'}^b \rangle\big] \leq \Theta\left(\frac{\sqrt{2\pi c}\left(\frac{c}{e}\right)^c}{2 n^{\frac{c}{2}} \sqrt{\pi c}\left(\frac{c}{2e}\right)^{\frac{c}{2}}}\right)$$

$$= \Theta\left(\frac{\left(\frac{c}{e}\right)^{\frac{c}{2}}}{n^{\frac{c}{2}}}\right)$$

$$= \Theta\left(\left(\frac{c}{en}\right)^{\frac{c}{2}}\right)$$

If $c = o(n)$ obviously $\frac{c}{en}$ aims to 0. If $c = \Theta(n)$, then $c$ also aims to $\infty$. Recall that $T \leq n$, which means that $c \leq 2n < en$, which means that $\left(\frac{c}{en}\right)^{\frac{c}{2}} \leq \frac{2}{e}^{\Theta(n)}$ aims to 0. ∎

**Theorem B.14 (Theorem 3.2)** *Any allocation $\mathcal{A}$ learning the encoder $B \in R^{n \times b}$ that follows both of the following conditions is maximal:*

1. *No row of B has more than Tm learnable weights*

2. *No columns of B has more than Td learnable weights*

*Every other allocation is minimal.*

**Proof:** To avoid confusions with number of steps, $T$, we use $\dagger$ to denote transpose matrix, e.g, $A^\dagger$. From Lemma A.3, Equation (3.3) is equivalent to:

$$\sum_{t=1}^{T} \left( X_t^\dagger \otimes DW^{T-t+1} \right) vec(B) = vec(Y)$$

For shortness, we denote $D_t = DW^{T-t+1}$ and $D_t^i$ its $i$th columns, and $X_t^i$ the $i$th row of $X_t$ (which is the $i$th column of $X_t^\dagger$).

We separate the columns of $\sum_{t=1}^{T} \left( X^\dagger \otimes DW^{T-t+1} \right)$ to two parts - the first part, denoted by $\tilde{C}$ is the columns that is multiplied by the learned entries of $B$, denoted by $\tilde{B}$. The other columns, denoted by $\widehat{C}$, are multiplied by the constant part of $B$, denoted by $\widehat{B}$. Thus:

$$\tilde{C}\tilde{B} = vec(Y) - \widehat{C}\widehat{B}$$

That means that the allocations matches iff $\tilde{C}$ is invertible. Since $r = dm$, $\tilde{C} \in \mathbb{R}^{dm \times dm}$, we can say the allocation matches iff the columns of $\tilde{C}$ are linear independent.

Every columns of kronecker product is a vectorization of outer product of two vectors. In our case, it means that the columns of $X_t^\dagger \otimes D_t$ are $X_t^i \otimes D_t^j$ for some $i, j$. Denote:

$$V_i = \left( \begin{array}{c|c|c|c|c} X_1^i & X_2^i & \cdots & X_{t-1}^i & X_T^i \end{array} \right)$$

$$U_i = \begin{pmatrix} D_1^i \\ D_2^i \\ \vdots \\ D_{T-1}^i \\ D_T^i \end{pmatrix}$$

The means that the the columns of $\tilde{C}$ is $vec(V_i U_j)$ for some $i, j$. Specifically, if the $(j, i)$ entry of $B$ is in the allocation $\mathcal{A}$, it means that $V_i U_j$ will be in $\tilde{C}$.

Thus, the allocation matches iff $\{V_i U_j\}_{(i,j \in \mathcal{A})}$ is linear independent, namely a basis for $\mathbb{R}^{m \times d}$. To use Lemma B.12, we need to show that there is no correlations. $V_i$ is built from the i.i.d $X$, so there should be no correlations, and there are not correlations in $U_i$ from Lemma B.13 and corollary B.11.

Since $V_i$ and $U_i$ has no internal correlations, from Lemma B.12 (with $k = T$) we get that $\{V_i U_j\}_{(i,j \in \mathcal{A})}$ is if $V_i$ is a basis iff the set doesn't contain the same $U_i$ more then $m$ times or the same $V_i$ more then $d$ times. Since the set contains the if the $(j, i)$ entry of $B$ is in the allocation $\mathcal{A}$, it means that $V_i U_j$ will be in the set, it will be a basis iff $\mathcal{A}$ follows exactly the conditions in the theorem. ∎

## C   Linear feed-forward

**Theorem C.1** *For any allocation $\mathcal{A}$ learning an intermediate layer $W_l \in \mathbb{R}^{n_l \times n_{l-1}}$ is maximal if and only if it follows one of the following:*

1. *There is no rows that has more than $m$ learnable weights*

2. *There is no columns that has more than $d$ learnable weights*

*Otherwise, the allocation is minimal.*

**Proof:** For any $1 \leq i \leq j \leq L$, denote $W_j W_{j-1} \cdots W_{i+1} W_i$ as $W_{j:i}$.

Assume that the learned matrix is the $l$-th layer, namely $W_l$. We can express the network as:

$$Y = W_{L:l+1} W_l W_{l-1:1} X$$

For brevity, denote $A = W_{L:l+1}$, $W = W_l$, and $B = W_{l-1:1} X$. Here, $A \in \mathbb{R}^{d \times n_{l+1}}$ and $B \in \mathbb{R}^{n_l \times m}$. From Lemma A.3, we have:

$$(B^T \otimes A) vec(W) = vec(Y)$$

Notice that every entry in $vec(W)$ corresponds to a column in $(B^T \otimes A)$. Denote $\tilde{W}$ as the learnable part of $W$ and $\widehat{W}$ as the constant part. Respectively, denote $\tilde{C}$ as the matrix created from the columns of $(B^T \otimes A)$ corresponding to $\tilde{W}$, and $\widehat{C}$ as the matrix from the columns corresponding to $\widehat{W}$. We can write:

$$\tilde{C} \tilde{W} = vec(Y) - \widehat{C} \widehat{W}$$

Since $r = dm$, we have $\tilde{C} \in \mathbb{R}^{dm \times dm}$. The equation is solvable if and only if $\tilde{C}$ is invertible.

Each different allocation corresponds to a distinct choice of $dm$ columns out of the $n_l n_{l+1}$ columns of $(B^T \otimes A)$. Every column of a Kronecker product is a vectorization of the outer product of two vectors. In our case, denote $A_i$ as the $i$-th column of $A$ and $B_i^T$ as the $i$-th column of $B^T$ (which is the $i$-th row of $B$). If the $(i, j)$-th entry of $W$ is included in the allocation, then the column created from the outer product of $A_i$ and $B_j^T$ will be part of $\tilde{C}$.

Since $\tilde{C}$ is a square matrix, it is invertible if and only if its columns are linearly independent. As the columns are vectorizations of matrices, they are linearly independent if the matrices form a basis for $\mathbb{R}^{d \times m}$. From Corollary B.11, both $A$ and $B$ are uncorrelated, allowing us to use Lemma B.12 (with $k = 1$) to determine if these matrices indeed form a basis. One can observe that the matrices will form a basis if and only if the conditions in this theorem hold.

■

## D SHALLOW RELU

**Lemma D.1** *Let $v \in \mathbb{R}^n$ and $P \in diag(\{0, 1\}^n)$. Then, $(2P - I)v \geq 0$ if and only if $\phi(v) = Pv$, where $\phi$ is the ReLU function.*

**Proof:** Let $P_i$ represent the $i$-th diagonal entry of $P$.

1. **First direction**: Assume $(2P - I)v \geq 0$. This implies that for every $1 \leq i \leq n$, $(2P_i - 1)v_i \geq 0$. When $P_i = 1$, it follows that $v_i \geq 0$, and when $P_i = 0$, we have $v_i < 0$. Therefore, $\phi(v_i) = P_i v_i$ holds for each $i$, and in general, $\phi(v) = Pv$.

2. **Second direction**: Now, assume $\phi(v) = Pv$. This means that $P_i = 1$ when $v_i \geq 0$, and $P_i = 0$ when $v_i < 0$. Hence, $(2P_i - 1)v_i \geq 0$ holds for all $i$, which implies $(2P - I)v \geq 0$. ■

**Lemma D.2** *Let $\mathcal{A}$ be an allocation scheme that assigns learnable weights to $k$ rows. Without loss of generality, assume it is the first $k$ rows.*

*Denote by $W_2^1$ the first $k$ columns of $W_2$ and by $W_2^2$ the remaining columns. Similarly, let $W_1^1$ be the first $k$ rows of $W_1$ and $W_1^2$ the remaining rows.*

*For a set of samples $X$ and labels $Y$ generated by a teacher model, a student model matches the teacher if, for each $1 \leq i \leq m$, the following system of equations and inequalities is solvable:*

$$W_2^1 P_i W_1^1 x_i = y_i - W_2^1 \phi(W_1^2 x_i) \tag{D.1}$$

$$(2P_i - I)W x_i \geq 0 \tag{D.2}$$

$$P_i \in diag(\{0, 1\}^k), \quad W_1 \in \mathbb{R}_{\mathcal{A}}$$

**Proof:** Notice that the label $y_i$ is given by:

$$y_i = W_2 \phi(W_1 x_i) = W_2^1 \phi(W_1^1 x_i) + W_2^1 \phi(W_1^2 x_i)$$

From Lemma D.1, this leads to the system of equations and inequalities described above, completing the proof. ∎

**Lemma D.3** *For any allocation scheme, the probability that a given $P_i$ has fewer than $d$ ones and still satisfies Lemma D.2 is zero.*

**Proof:** Write $P = P_i$ for brevity. Suppose $P$ has fewer than $d$ ones. Then the rank of $W_2 P$ is less than $d$. If the equations from Lemma D.2 are solvable, we can write:

$$(x_i^T \otimes W_2^1 P) \operatorname{vec}(W_1^1) = y_i - W_2^1 \phi(W_1^2 x_i)$$

Since $(x_i^T \otimes W_2^1 P) \in \mathbb{R}^{d \times qn}$, and by Lemma A.4, the rank is less than $d$, any matrix formed from a subset of $d$ columns will be invertible with probability zero. Therefore, the equation is solvable with probability zero. ∎

This means that for every $P_i$, there are $\sum_{i=0}^{d-1} \binom{k}{i}$ configurations for $P_i$ that are not usable. Notice that $2^k = (1+1)^k = \sum_{i=0}^{k} \binom{k}{i}$, which shows that as $k$ grows larger, the fraction of configurations with zero probability decreases. For example, when $k = d$, there is only one valid configuration for $P$, as $2^d - \sum_{i=0}^{d-1} \binom{d}{i} = 1$.

**Lemma D.4** *Let $\mathcal{A}$ be an allocation scheme that allocates $r_i$ learnable weights in the $i$-th row. Let $P_{ji}$ represent the value in the $i$-th entry on the diagonal of $P_j$. If the number of ones in $\{P_{li}\}_{l=1}^{m}$ is less than $r_i$, then the probability that the system satisfies Lemma D.2 is zero.*

**Proof:** For brevity, let the column vectors of $W_2^1$ be $\{a_i \in \mathbb{R}^d\}_{i=1}^{k}$. If the equations in Lemma D.2 are solvable, we can write:

$$(x_i^T \otimes W_2^1 P_i) \operatorname{vec}(W_1^1) = y_i - W_2^1 \phi(W_1^2 x_i)$$

for $1 \le i \le m$. These represent $dm$ linear equations.

Denote $v_{ij} \in \mathbb{R}^m$ as $\{X_{lj} P_{li}\}_{l=1}^{m}$, where $X_{lj}$ is the $j$-th entry of the $l$-th sample and $P_{li}$ is the $i$-th diagonal element of $P_l$. In the matrix of this linear equation, the columns corresponding to index $i$, $j$ in $vec(W_1^1)$ are the vectorizations of the outer products $v_{ij} \otimes a_i$. Since we have $dm$ equations with $r = dm$ variables, those columns must be linear independent for the equations to be solvable.

If $r_i$ columns corresponding to index $i$ are selected, linear independence must hold between the matrices $v_{ij} \otimes a_i$ for $r_i$ choices of $j$. However, if the number of ones in $\{P_{li}\}_{l=1}^{m}$ is less than $r_i$, it means that each vector $v_{ij}$ has more than $m - r_i$ zero entries, which means that any set of $r_i$ options for $j$, the vectors $v_{ij}$ will be linear dependent. Therefore, the matrices $v_{ij} \otimes a_i$ will also be linear dependent, which means that the equation is not solvable. ∎

Notice that out of the $2^{km}$ possible configurations for $P$, only $\prod_{i=1}^{k} \sum_{j=r_i}^{m} \binom{m}{j}$ are feasible. This shows that allocations using more rows (i.e., larger $k$) and stacking fewer learnable weights per row (since the sum starts from $r_i$) have significantly more possible configurations.

# E EXPERIMENTS

## E.1 METHODS

As we have seen multiple times in this paper, the match probability often boils down to the likelihood of a set of polynomials having an exact solution. Standard gradient descent failed to find a solution in this case. Typically, gradient descent can overfit a dataset because the model is often overparameterized relative to the data. However, in our case, the model isn't overparameterized (as we use $r = dm$), so second-order optimization methods were necessary.

Instead of gradient descent methods, we used Python's `scipy.optimize.fsolve`. For each experiment, we ran the solver 1,000 times, and the match probability was calculated based on the number of solutions found. Since we used second order optimization, we were limited to use only small network sizes. However, the used networks sizes was enough to show the expected phenomena.

Each run of `fsolve` begins with an initialization point $x_0$ for the algorithm. Empirically, we observed that `fsolve` frequently fails to find a solution initially, but with multiple initializations, it eventually succeeds. Across all experiments, we identified a threshold of initializations that, after being surpassed, rarely leads to the discovery of additional solutions. We manually evaluated this threshold for each experiment and then doubled that number. For instance, in the LRNN experiments, we observed that no new solutions were found after 200 initializations, so we set the number of initializations to 400 to ensure thoroughness.

In the ReLU experiment, we did not use `fsolve` as it failed to find any solutions in all cases. Instead, we employed Ada Hessian (Yao et al., 2021), and considered a solution valid if the mean squared error (MSE) was below a threshold of $10^{-2}$. Additionally, we recorded the number of solutions found across 1,000 attempts, and the resulting figure includes a shaded area to represent one standard error of the mean over these trials.

The experiment on MNIST data was the only one where we employed first-order optimization, specifically using the Adam algorithm (Kingma and Ba, 2017). This approach allowed us to utilize a larger network with $n = 1000$ parameters. We conjecture that first-order optimization performed better in this case due to two factors: (1) random data is inherently more challenging to learn as it lacks any underlying correlations, and (2) in the MNIST experiment, the task was to classify images (where the network predicts a label between 1-10) rather than reproducing a full vector, which simplifies the learning objective.

### E.2 ENVIRONMENT

**Figure 2** Figure 2 was created with $T = \frac{n}{2}$, $b = 1, d = 4$, and $m = \frac{n}{2}$. All random variables were drawn from a normal distribution with normalized variance (see Appendix E.3). Let the number of used rows be denoted as $k$. For every possible $k$, we ran 1,000 trials and averaged the number of matches to estimate the match probability.

For each trial with a given $k$, an allocation was randomized in the first $k$ rows, subject to the conditions in Theorem 3.4. Note that Theorem 3.4 identifies the minimal number of rows for which below it there is no solution for any allocations ($k < d$; minimal allocations). This is why the graph doesn't start at $k = 0$.

In Figure 2b, $d = \frac{n}{4}$. In this scenario, when $k = d$ (0.25 on the x-axis), the only allocations that satisfy Theorem 3.4 are those that also satisfy Theorem 3.5. As expected, these allocations have a match probability of 1. Since the proof of Theorem 3.5 provides an algorithm to find a solution in this case, we used that algorithm instead of `scipy.solve`. The algorithm is included in the `maximality_lrnn.py` file in the attached zip.

**Figure 3** The feedforward network in Figure 3a was created with three layers, with $q = 4$, $d = 6$, and $m = 4$. The two hidden layer has the same size, denoted by $n$ in the graph. This means that there were $r = md = 24$ learnable weights, distributed as 8 per layer. Just like in the LRNN, the allocation was randomized across the rows, adhering to the conditions specified in Theorem 4.1. Notably, for the first layer, the number of rows does not impact the number of linear equations in Equation (4.2). Therefore, we set $k$ as a limit for the number of columns utilized.

The ReLU network in Figure 3b was created with $q = 6$, $d = 4$, and $m = 6$. The size of the hidden layer, denoted by $n$, is provided in the graph.

The experiment on MNIST data shown in Figure 3c was conducted using a network with $n = 1000$ and $m = 1000$, while the input and output sizes were set to $q = 784$ and $d = 10$, respectively, reflecting the dimensions of the MNIST dataset.

### E.3 Variance in the experiments

Let $v \in \mathbb{R}^n$ and $h \in \mathbb{R}^n$ be two random vectors, where $v \in \mathbb{R}^n$ drawn i.i.d from $\mathcal{N}(0, g^2)$ and $h \in \mathbb{R}^n = \Theta(1)$. Since they are uncorrelated, $\mathbb{E}[\langle v, h \rangle] = 0$. However, the variance of their inner product is:

$$\mathbb{E}\left[\langle v, h \rangle^2\right] = \mathbb{E}\left[\sum_{i=1}^{n} v_i^2 h_i^2\right] = \sum_{i=1}^{n} \mathbb{E}\left[v_i^2 h_i^2\right] = \sum_{i=1}^{n} \mathbb{E}\left[v_i^2\right]\mathbb{E}\left[h_i^2\right] \propto ng^2 = \Theta(n)$$

Therefore, networks that multiply matrices repeatedly this way (e.g., $Wh_t$, $Dh_t$) will cause the variance of the random variables to explode.

The solution is to sample all matrices with row size $n$ with variance $\frac{g^2}{n}$, with $g = \Theta(1)$. This ensures that the variance of the hidden state remains $\Theta(1)$ throughout the process.

## F Supplementary Discussion

A natural point of comparison for our work is the Lottery Ticket Hypothesis (LTH) framework, which focuses on identifying sparse subnetworks within over-parameterized models that, when trained independently, achieve comparable performance to the full network. A key distinction between our work and LTH lies in the nature of the problem addressed and the context of sparsity. LTH focuses on identifying "winning tickets" — sparse subnetworks within an already over-parameterized model—that achieve comparable performance to the full network when trained independently. In contrast, our work examines how to strategically allocate a fixed, limited number of learnable weights across a network to maximize its expressivity. This difference is critical: while LTH emphasizes *discovering* useful sparsity post hoc, our approach is about *designing* useful sparsity under strict resource constraints from the outset.

The student-teacher setup is a well-established framework for studying machine learning problems in controlled settings. It has been widely used in the literature to analyze generalization, expressivity, and optimization (e.g., see (Saglietti et al., 2022), which also includes a comprehensive set of references). The choice of a student-teacher setup in our work is deliberate and is done for clarity, as it isolates the reduction in expressive power arising solely from the allocation of learnable weights, rather than confounding factors such as differences in architecture or neuron nonlinearities. Specifically, when the teacher and student share the same architecture, any decrease in the student's expressive power is attributable solely to the restriction in learned weights and their allocation. This allows us to rigorously estimate the approximation error stemming from allocation strategies, independent of other factors that might limit the student's ability to fit the labeled data. While we used the student-teacher setup for clarity, the framework is not inherently limited to this context. In fact, the teacher and student could differ in architecture, and the analysis could extend to general labeled data (the proofs in this paper remain valid under these conditions).

With respect to applications in neuroscience, little is known about the scale of learning (i.e., changes in synaptic weights) in the brain. Historically, technological constraints have made it notoriously difficult to track synaptic weight changes in real-time as animals learn new tasks (Tsutsumi and Hayashi-Takagi, 2021). However, it is well established that learning induces changes in neural activity that are often highly distributed both within and across brain regions (Chen et al., 2017; Steinmetz et al., 2019; Allen et al., 2019). Notably, a recent study demonstrated that even when learning is localized to a subset of neurons and synaptic weights, the resulting activity can propagate through fixed-weight connections, leading to widespread changes in neural activity Kim et al. (2023). This observation highlights the challenge of relating widespread neural activity to the specific extent of synaptic weight changes in the brain.

Recent technological advances now enable neuroscientists to monitor changes in synaptic weights during learning, offering unprecedented insights into large-scale connectivity dynamics Daie et al. (2021); Humphreys et al. (2022); Finkelstein et al. (2023). If the findings from our theoretical study extend to more complex network architectures and neuron nonlinearities, these innovations could make our predictions and insights testable in the near future. This possibility served as one of the motivations for conducting this research.

### F.1 Assumptions Relaxation

In this paper, we assumed that $r = md$ (see Section 2). In the linear models we considered, this assumption is reasonable, as it ensures that the number of equations matches the number of free variables. As a result, $r = md$ is the minimal number of learnable weights needed to match the teacher. This allowed us to establish "sufficient conditions" theorems (Theorems 3.1, 3.2, 3.5 and 4.1) to guarantee that no learnable weight is "wasted". The reasoning is straightforward—since all learnable weights are generally needed to solve the equations, any excess usage in one subset of equations would lead to a shortage elsewhere.

Consequently, if $r > md$, up to $r - md$ learnable weights can be "wasted" without violating these conditions. This follows from the fact that when the number of free variables exceeds the number of constraints, there is additional flexibility in weight allocation, allowing some weights to be used inefficiently while still maintaining solvability.

One of the assumptions in this paper is that the input distribution $\mathcal{X}$ is such that any $m$ samples drawn from it are linearly independent. In our linear models, if the samples are linearly dependent, the effective number of distinct samples is reduced. Specifically, for any set of $m$ samples, there exists a subset of $m'$ independent samples (with $m' \leq m$) such that a match on these $m'$ samples implies a match on all $m$ samples. Therefore, we can treat the effective sample size as $m'$ and assume $m = m'$ in our analysis. This reinforces our assumption that, in the linear case, the samples can be considered effectively linearly independent.

