# OpenReview forum: "The impact of allocation strategies in subset learning on the expressive power of neural networks"
_ICLR.cc/2025/Conference — ICLR 2025 Poster_

### Official Review · Reviewer_HB5t · 2024-10-26

**Soundness:** 4
**Presentation:** 4
**Contribution:** 4
**Rating:** 8
**Confidence:** 4

**Summary:**

The authors investigate the capacity of neural networks under constrained conditions where only a fixed number of weights are plastic and can be changed. The authors present theoretical work and then relate the theory to empirical simulations in simple networks.
This work is quite interesting in that it forces us to think that not all weights are equal, that not all weights are needed, and that astute ways of choosing which weights to change can have a large impact in the capacity and efficacy of a neural network.

**Strengths:**

This work introduces an interesting question about the possibility to modify only certain weights in a network and ask how the choice of plastic weights impacts neural network capacity.
There has not been much work along these lines. Perhaps one related idea relates to continual learning. Upon learning new things, it may be useful to strategically decide which weights to change more. One instantiation of this is Kirkpatrick et al PNAS 2017 (the authors cite other work in this direction as well).
The work presents both theory and empirical simulations.
The work introduces a benchmark that would be useful for the field to compare different potential mechanisms to allocate learnable parameters.

**Weaknesses:**

The abstract states that “biological systems like the brain where evidence suggests that only a subset of synaptic weights are modified during learning”. This is a grandiose claim that is not substantiated by any references. I strongly suggest that they remove this claim. Otherwise, the authors should provide compelling evidence about this statement. The introduction provides a better assessment: “the scale of this process in the brain is unclear”.
The experiments presented are based on rather simple NNs. It is not clear how the conclusions extrapolate to more complex NNs. But this point does not distract from the beauty of the work presented here.

**Questions:**

The potential connections between the work presented here and weight protection strategies in continual learning is probably worth exploring further (though this is not a requirement by any means for the current work).

---

> ### Author Response · Authors · 2024-11-19
> **Detailed reply to reviewer HB5t**
>
> We thank the reviewer for their comments. We went over the points raised by the reviewer and addressed them as detailed in the following text.
>
>
> **Comment**: *This is a grandiose claim that is not substantiated by any references. I strongly suggest that they remove this claim. Otherwise, the authors should provide compelling evidence about this statement. The introduction provides a better assessment: “the scale of this process in the brain is unclear”*.
>
>
> **Response**: While there is some evidence to this claim, and references were provided in the introduction, following the reviewer’s suggestion, we removed this claim from the abstract, and changed it accordingly.
>
>
> **Comment**: *The experiments presented are based on rather simple NNs. It is not clear how the conclusions extrapolate to more complex NNs. But this point does not distract from the beauty of the work presented here.*
>
>
> **Response**: We appreciate the reviewer’s thoughtful feedback and are grateful for their recognition of the beauty of the work, which we believe stems from the novel setting, problem formulation, and the theoretical analysis and insights presented. The primary goal of this paper is theoretical, and the experiments were designed to support the theoretical results. Consequently, the neural networks used in our experiments are intentionally kept simple to directly connect the results to our theoretical framework.
>
> We have now included additional results using the MNIST dataset, which demonstrate that our conclusions also hold for structured, real-world data, that goes beyond our theory. While these results are a promising first step, we agree that future work should explore allocation strategies within our framework using more complex networks. Yet, given the generality of our theoretical setting (see response to mi9c), we are optimistic that our insights can be at least tested numerically on more intricate architectures in future studies.
>
> **Comment**: *The potential connections between the work presented here and weight protection strategies in continual learning is probably worth exploring further (though this is not a requirement by any means for the current work).*
>
> **Response**: We thank the reviewer for highlighting the potential connections between our work and weight protection strategies in continual learning. As mentioned in the introduction, there is indeed a conceptual relationship between continual learning and subset learning.
>
> In this paper, we chose to focus on the theoretical analysis of how different allocation strategies influence the expressive power of networks. This decision allowed us to thoroughly investigate and establish foundational insights within this specific context (see also our response to another reviewer regarding the scope of the paper).
>
> We agree that exploring allocation strategies in the context of continual learning, such as optimizing weight allocation to mitigate catastrophic forgetting, is a compelling direction for future research. It is beyond the scope of the current study.

---

> > ### Comment · Reviewer_HB5t · 2024-11-23
> > **Great theoretical work**
> >
> > The responses to my previous queries make sense to me. I am excited about this work.

---

### Official Review · Reviewer_QyMh · 2024-10-28

**Soundness:** 3
**Presentation:** 3
**Contribution:** 2
**Rating:** 6
**Confidence:** 4

**Summary:**

This paper provides a theoretical investigation into the distribution of learnable weights in neural networks, drawing inspiration from brain mechanisms. The study measures network expressivity as a function of these learnable weights, examining various network architectures and validating the theory through experiments.

**Strengths:**

Congratulations on your submission to ICLR! I commend the authors on producing this great piece of research. Below is my detailed review of the work:

**Originality**
This paper introduces a theoretical framework for analyzing network expressivity as a function of learnable weight allocation. Its originality lies in drawing parallels with brain mechanisms and adopting a fresh theoretical approach.

**Quality**
The paper demonstrates high quality with well-substantiated theoretical findings that are clearly explained and motivated. Key strengths include:
- Thorough and detailed theoretical results, effectively show that predictions from smaller models generalize well to non-linear, shallow ReLU architectures.
- Theoretical analysis is rigorously structured, adding robustness to the findings.
- The mathematical derivations appear accurate and sound.

**Clarity**
The paper is well-organized and accessible.

- The use of a simplified linear network model as a toy example helps to build intuition, significantly enhancing readability.
- The figures are clear, informative, and effectively support the main content.

**Significance**
This work is a valuable contribution to the machine learning research community, especially in deepening our understanding of model expressivity as a function of trainable weights in the context of increasing training costs. Additionally, the proposed theoretical framework may offer valuable insights for the neuroscience community.

**Weaknesses:**

**Originality**
No notable weaknesses in originality.

**Quality:**
The paper is of high quality, yet there are areas where further clarity and detail could strengthen its impact:

- The paper would benefit from a more explicit discussion of its limitations. While valuable, the theoretical approach may not fully capture some empirical phenomena seen in practical neural networks. Highlighting these potential divergences between theory and real-world observations would add useful nuance.
- A clearer connection to applications in neuroscience would enhance the paper’s relevance to that field.

**Clarity**
The clarity of the paper is generally high but could be improved in specific areas:

- An introductory figure would be helpful, particularly for readers unfamiliar with the topic. For instance, a visual summary of the student-teacher setup could make the content more accessible.
- Including a discussion in the introduction on the growing computational demands—now requiring substantial resources—would add timely context to the study.
- Strengthening the link between neural networks and brain mechanisms would clarify the relevance of the approach.
- Developing Theorem 3.3 further, potentially in the appendix, would also improve clarity.

**Significance**
The significance of the study could be enhanced with a few adjustments:
- The benchmarks used primarily involve toy models, which may limit the generalizability of the findings. Extending the analysis to larger models could increase the impact and relevance of the results.

**Questions:**

-  Are there any constraints preventing from running the experiments with gradient descent? Is it correct to think that if you did you could have access to results with larger networks?
- Do you have any intuition on how these insights might scale to more complex networks? Specifically, if larger models contain an increasing number of polynomials increases the probability to find a solution , does this imply that weight distribution becomes less significant, given that the transition would be so abrupt that the network would almost never (with a probability tending to zero) have no solution or expressivity zero ?
- (This may be outside the intended scope, but I’m curious.) I understand that the approach abstracts away the specifics of the learning algorithm. However, I’m interested in how your findings might depend on the network’s initialization under a particular learning algorithm (e.g., the scale and relative scaling of layers). Given that different initializations can influence the learning rates of certain weights more than others—effectively controlling which parts of the network learn more actively—how does the network’s expressivity vary under these conditions?

I hope this review is helpful for the further development of the paper. I encourage the author to continue this research and incorporate the feedback provided.

---

> ### Author Response · Authors · 2024-11-19
> **Detailed reply to reviewer QyMh (1/2)**
>
> We thank the reviewer for their comments. We went over the points raised by the reviewer and addressed them as detailed in the following text.
>
> **Comment**:  *The paper is of high quality, yet there are areas where further clarity and detail could strengthen its impact:*
> * *The paper would benefit from a more explicit discussion of its limitations. While valuable, the theoretical approach may not fully capture some empirical phenomena seen in practical neural networks. Highlighting these potential divergences between theory and real-world observations would add useful nuance.*
> * *A clearer connection to applications in neuroscience would enhance the paper’s relevance to that field.*
>
> **Response**: We thank the reviewer for highlighting the quality of the paper. We now added to the discussion a paragraph on the limitations of our framework.. In addition, we added a paragraph in the supplementary discussion on the relevance of the paper to neuroscience.
>
> **Comment**: *The clarity of the paper is generally high but could be improved in specific areas:*
> * *An introductory figure would be helpful, particularly for readers unfamiliar with the topic. For instance, a visual summary of the student-teacher setup could make the content more accessible.*
> * *Including a discussion in the introduction on the growing computational demands—now requiring substantial resources—would add timely context to the study.*
> * *Strengthening the link between neural networks and brain mechanisms would clarify the relevance of the approach.*
> * *Developing Theorem 3.3 further, potentially in the appendix, would also improve clarity.*
>
> **Response**: We thank the reviewer for appreciating the clarity of the paper and their suggestions to improve it. We included an introductory figure and mentioned the growing computational demands as a motivation for this study. In addition, we added a paragraph  on the relevance of the paper to neuroscience, and elaborated on Theorem 3.3 (now Theorem 3.2).
>
> **Comment**: *The benchmarks used primarily involve toy models, which may limit the generalizability of the findings. Extending the analysis to larger models could increase the impact and relevance of the results.*
>
> **Response**: We appreciate the reviewer’s comment regarding the benchmarks used in our study. Toy models are a well-established approach for advancing theoretical work, as they provide a simplified yet controlled setting to develop and validate foundational insights. In response to the reviewer’s feedback, we have expanded our analysis to include numerical simulations using the MNIST dataset. These simulations demonstrate that our findings extend beyond toy models and apply to more realistic, structured datasets. With respect to larger models, please refer to our responses below and to other reviewers.
>
> **Comment**: *Are there any constraints preventing from running the experiments with gradient descent?*
>
> **Response**: In our simulations, we trained the networks to reproduce random data, which aligns closely with the theoretical analysis we presented in the paper. Empirically, we found that gradient descent (GD) performed poorly, even on very small networks, on random data with a limited number of learnable weights. Notably, GD is known to perform well on convex functions (see Boyd & Vandenberghe, Chapter 9.3) and on certain over-parameterized, non-convex functions, as demonstrated in recent research (e.g., Allen-Zhu et al., 2018), but neither case applies to our work. Thus, we opted to use second-order optimization methods, which worked much better in expressing the teacher’s data. Unfortunately, these methods scaled poorly for larger networks.
>
> However, we anticipate that GD methods could work better on structured data. Indeed, we note that when we estimated the Match Probability (MP) in new simulations using MNIST data we could scale our simulations to much larger networks. These results aligned well with our theoretical insights.

---

> ### Author Response · Authors · 2024-11-19
> **Detailed reply to reviewer QyMh (2/2)**
>
> **Comment**: *Do you have any intuition on how these insights might scale to more complex networks? Specifically, if larger models contain an increasing number of polynomials increases the probability to find a solution, does this imply that weight distribution becomes less significant, given that the transition would be so abrupt that the network would almost never (with a probability tending to zero) have no solution or expressivity zero?*
>
> **Response**: Our findings show that, at least for linear networks, distributing the learnable weights increases the likelihood of an allocation being maximal. These intuitions apply to both deep and recurrent linear networks. We expect that the logic underlying our theoretical analysis, which led us to these insights, will also be valid for any linear network, including more complex architectures of linear neurons. Additionally, we demonstrated that similar principles hold for shallow nonlinear networks. The combination of these observations suggests that the principle of distributing learnable weights to increase expressivity can extend to more complex nonlinear networks. Future work is needed to validate whether our intuitions are correct and to identify the circumstances under which they may fail.
>
> Regarding the reviewer’s comment, we agree that in larger models, the distribution of learnable weights may become less critical as the number of polynomials increases—provided the weights are sufficiently distributed so that allocations are not minimal. This observation aligns with the sharp transition seen in the figures and reflects our intuition, which is based on counting the number of polynomials. In fact, this phenomenon appears also in the new simulations we ran on the MNIST data (Figure 3c).
>
> However, further research is necessary to rigorously explore this scaling behavior, for several reasons. First, our findings suggest that the significance of weight distribution depends on whether the architecture leads to a system of nonlinear equations. For instance, training multiple layers or recurrent connections in LRNNs (as opposed to only training the encoder or decoder) results in such equations, while training restricted to the encoder, decoder, or a single layer in a deep network does not. Thus, simply increasing network size is not sufficient to guarantee an increase in the number of polynomials. Second, our conjecture that a higher number of polynomials facilitates finding solutions is supported by simulations and theoretical results (e.g., Subag, 2024). These theoretical results assume specific conditions, such as Gaussianity of the polynomial coefficients. While our simulations have not identified deviations from this trend, it is possible that certain structural properties of the model architecture could lead to contrary outcomes. In summary, while our work provides strong initial evidence and intuition, a deeper understanding of how these principles scale to larger and more complex networks requires further study.
>
> **Comment**: *Given that different initializations can influence the learning rates of certain weights more than others—effectively controlling which parts of the network learn more actively—how does the network’s expressivity vary under these conditions?*
>
> **Response**: This is a compelling question, and we agree that investigating how varying learning rates might impact expressivity could provide additional insights into weight allocation. We suggest that with multiple learning rates, the faster weights could act as a subset, learning relative to the slower (quenched) weights in a way that resembles a form of allocation. This layered learning speed is indeed a heuristic that might parallel our concept of allocation, though further theory would be required to fully understand this dynamic, which leads to a hierarchy of allocations.

---

> > ### Comment · Reviewer_QyMh · 2024-11-23
> > **Thank you for the detailed rebuttal**
> >
> > I commend the authors' efforts in providing clear explanations, expanding on the applications, and offering detailed responses to the questions raised. I found the study of allocations in models trained on MNIST data and the paragraph addressing the framework's limitations particularly insightful.
> >
> > However, I believe two significant limitations remain in this work. First, gradient descent (GD), one of the most widely used algorithms in standard machine learning, was observed to perform poorly in this context. While it is unclear which learning method the brain employs, there is some speculative potential for applications in this area, though this connection remains highly uncertain.
> >
> > Second, the diminishing importance of the distribution of learnable weights as the number of polynomials increases makes this theory challenging to reconcile with the use of neural networks in practice.
> >
> > While I strongly support the use of toy models, acknowledging their value as a proven and effective approach to advancing theoretical work, these two limitations make it difficult to fully evaluate the broader impact of this study.
> >
> > Finally, I would like to point out that I found the discussion on the influence of learning rates particularly intriguing. I encourage the authors to explore this further, as it could add a new dimension to their work. Furthermore, l think this work could gain strength by being more closely related to neuroscience findings as developed in the answer of mi9c (which was also very interesting).
> >
> > Given these considerations, I am currently maintaining my overall score.

---

> > > ### Author Response · Authors · 2024-11-24
> > > **Detailed reply to reviewer QyMH (1/2)**
> > >
> > > **Comment**: First, gradient descent (GD), one of the most widely used algorithms in standard machine learning, was observed to perform poorly in this context. While it is unclear which learning method the brain employs, there is some speculative potential for applications in this area, though this connection remains highly uncertain.
> > >
> > > **Response**: We appreciate the reviewer’s feedback and would like to address the points raised. However, we found the comment somewhat ambiguous. Based on our understanding, the reviewer raised two concerns:
> > >
> > > 1. If gradient descent (GD) does not scale well, how does the brain achieve learning?
> > >
> > > 2. If GD does not scale well, how can allocations be studied in larger networks and more complex settings?
> > >
> > > We will address both points below.
> > >
> > > 1. Our work primarily focuses on the expressivity of neural networks rather than their optimization. As described in the manuscript, the theoretical results we present establish the existence of maximal allocations for large networks. However, developing algorithms to find such allocations is a distinct question, and our study does not address this directly. In small networks with random data, Hessian-based methods performed well in our experiments, as shown in the manuscript.
> > >
> > >    The question of how the brain learns is a longstanding and active area of research in neuroscience. While there is substantial knowledge about possible learning rules in the brain (Magee & Grienberger, 2020), tracking changes in synaptic weights in behaving animals has historically been challenging. This limitation has constrained our understanding of how optimization occurs in the brain, although recent advances in experimental neuroscience are beginning to address this gap (see supplementary discussion in the paper). In fact, the lack of knowledge about the actual learning rules in the brain motivated us to focus on studying expressivity rather than investigating allocation strategies under a specific learning rule.
> > >
> > >    Regarding gradient descent (GD), it is widely believed that standard GD methods are unlikely to represent the mechanisms employed by the brain (e.g., Crick, 1989). GD-inspired alternatives, such as feedback alignment and reinforcement learning-based methods like three-factor rules (Magee & Grienberger, 2020) and REINFORCE (Williams, 1992), have been proposed as biologically plausible mechanisms. However, these methods also tend to scale poorly in large networks (Lillicrap et al., 2022). Given this, it is unclear what the benefit would be, at this stage, of studying how specific learning rules can be used to optimally allocate learnable weights without knowing which rules the brain employs. For us, focusing on expressivity was a better option.
> > >
> > > 2. We acknowledge that GD performed poorly on random data in our experiments. However, as noted in our previous response, GD performed very well on MNIST, even in larger networks (n=1000, Fig. 3C). This result is promising as it demonstrates that widely used optimizers, such as ADAM, can successfully optimize large networks in the context of evaluating the match probability. These findings suggest that match probability (MP) can be estimated using GD in practical settings, even for more complex models. To strengthen this claim, we plan to include additional experiments in the camera-ready version. Specifically, we will focus on larger recurrent neural networks (RNNs), which are more directly relevant to modeling neural dynamics in the brain.

---

> > > > ### Author Response · Authors · 2024-11-24
> > > > **Detailed reply to reviewer QyMH (2/2)**
> > > >
> > > > **Comment**: Second, the diminishing importance of the distribution of learnable weights as the number of polynomials increases makes this theory challenging to reconcile with the use of neural networks in practice.
> > > >
> > > > **Response**: We appreciate the reviewer’s comment and view this as a continuation of the earlier discussion. We agree that the specific allocation of weights becomes less critical in large networks, provided the weights are sufficiently distributed to avoid minimal allocations. However, it is unclear why the reviewer finds this challenging to reconcile with the use of neural networks in practice. We would appreciate further elaboration on this point, especially considering that our work focuses on expressivity rather than generalization. Nevertheless, along these lines, several important points should be noted:
> > > >
> > > > First, even if weight allocation becomes less critical in large networks, this remains a non-trivial and surprising theoretical result. For example, the student can express the teacher even when each neuron has only one learnable weight. Our theory provides an explanation for this phenomenon, highlighting its underlying principles.In addition, it provides.
> > > >
> > > > Second, while our theory reveals that many weight distributions can achieve maximal allocations in large networks, there are still non-trivial constraints that must be satisfied to avoid minimal allocations. For instance, even in large networks, if all learnable weights are distributed across many neurons (rows) but concentrated in the output of a single neuron (i.e., one column is learned), the allocation, despite being row-distributed, becomes minimal (Theorem 3.4). This illustrates that distinguishing between maximal and minimal allocations is not always straightforward, even when the models are large. In more complex networks, additional constraints and subtleties are likely to emerge, making this an exciting direction for future research to further understand the interplay between allocation strategies and expressivity.
> > > >
> > > > **Comment**: Finally, I would like to point out that I found the discussion on the influence of learning rates particularly intriguing. I encourage the authors to explore this further, as it could add a new dimension to their work.
> > > >
> > > > **Response**: Investigating learning rates would involve delving into optimization dynamics, which requires a separate framework and different measurements, as the Match Probability (MP) metric is specifically designed to assess expressiveness rather than optimization. While we agree that this could add a valuable new dimension to the study, we emphasize that it is beyond the scope of our current work, which focuses on expressivity rather than optimization.
> > > >
> > > > **Comment**: Furthermore, l think this work could gain strength by being more closely related to neuroscience findings as developed in the answer of mi9c (which was also very interesting).
> > > >
> > > > **Response**: We agree. We added this to the revised manuscript (see supplementary discussion).

---

> ### Comment · Reviewer_QyMh · 2024-11-29
>
> Dear Authors,
> I would like to thank the authors for providing a clear and well-articulated response. Apologies for the lack of clarity in my previous comment.
>
> While I understand that this work focuses on expressivity, I find it difficult to fully separate it from the aspect of optimization. The authors have noted that gradient descent (GD) performs poorly in this context, which I see as a limitation. However, I acknowledge that the work on MNIST provides a partial response to this concern.
>
> Regarding the comment on the neuroscience literature, my intention was not to ask, "If gradient descent (GD) does not scale well, how does the brain achieve learning?" Instead, I aimed to highlight that GD may not necessarily pose a problem, as it remains unclear which specific rule the brain employs for learning. However, this lack of clarity also means that any connections drawn between the theory and its application are inherently more speculative (but interesting).
>
> Finally, I meant to highlight that neural networks often operate in regimes where weight allocation becomes less critical. This theory, therefore, might be limited to settings that are less common in practice. Additionally, most weight distributions used in practice tend to avoid failure cases, even though identifying "minimal allocations" may not always be straightforward.
>
> I want to highlight that my last two suggestions do not directly impact the score, as they are intended as recommendations for future work and fall outside the current scope of this paper ( as agreed by the authors).
>
> Thank you for your efforts and for engaging with this feedback. I have raised my score in light of the authors' thorough rebuttal.

---

### Official Review · Reviewer_mi9c · 2024-11-02

**Soundness:** 2
**Presentation:** 2
**Contribution:** 2
**Rating:** 3
**Confidence:** 4

**Summary:**

This paper develops a theoretical framework for analyzing the expressive power of neural networks when only allowing a subset of the network to learn. The authors focus on a student-teacher setup and defines expressivity as the likelihood that the student can replicate the teacher’s outputs. Using this metric, this work analyzed linear RNNs and deep linear feedforward networks, and evaluated the expressive power of different allocation strategies, then extended their analysis to one-layer ReLU feedforward networks.

**Strengths:**

The paper presents several theoretical results about the expressive power of linear networks when allowing a subset of the network to learn, and concludes that spreading the learnable weights in different parts of the network allows for stronger expressive power.

**Weaknesses:**

1.	There are many ways to evaluate allocation strategies, including generalization performance, capabilities of transfer learning and continual learning, etc. These are the interesting problems that the authors used to motivate their work. However, this paper particularly focuses on how allocation strategy affects the expressivity of a network, in a very specific setting, limiting the scope and potential insights that could be obtained from this work.
2.	The result of the work is restricted to a student-teacher framework in the sense that the central definition of the match probability is defined based on a student-teacher framework, where the student and the teacher have matching architectures, making it difficult to extend the results to real-world applications even heuristically. The numerical results shown in this paper is also limited to this synthetic task only.
3.	Various assumptions are dispersed and showed up in various places within the text, it would be better if the authors could explicitly list their assumptions when stating the theorems.
4.	I did not check all proofs carefully but multiple proofs in this paper is written in a very informal way, making it difficult to judge the validity of the proofs.

**Questions:**

1.	Both the student and the teacher weights are drawn from i.i.d. Gaussian, have you considered correlated weights in the teacher, or introducing structures into the teacher weights?
2.	The central result that it is better to allocate the learnable weights throughout the network. Is this consistent with existing findings about learning in the brain?
3.	When stating that “the probability of finding at least one solution increases with the number of polynomials”, does the degree of the polynomials play a role?
4.	In the proof of theorem B1, do you assume that W is diagonalizable? The sentence “the eigenvector xxx is a space” does not make sense to me. Furthermore, I do not see why it is important to prevent the network from being degenerate, it is still a valid function of the input sequence, and it’s simply that the degree of freedom is less than the amount of effective linear weights.

---

> ### Author Response · Authors · 2024-11-19
> **Detailed reply to reviewer mi9c (1/3)**
>
> We thank the reviewer for their comments. We went over the points raised by the reviewer and addressed them as detailed below.
>
> **Comment**: *There are many ways to evaluate allocation strategies, including generalization performance, capabilities of transfer learning and continual learning, etc... This paper particularly focuses on how allocation strategy affects the expressivity of a network*
>
> **Response**: Our work raises a novel problem, on how different allocations affect the performance of the model. We agree with the reviewer that there are multiple ways to evaluate ‘performance’, and as an initial step to explore this general problem we focused on expressivity as a key metric. Numerous works studied expressivity of neural networks as a way to evaluate model capacity and has led to significant insights in various studies on neural networks (see references in the paper). Specifically, it was used before as a measurement to compare between different architectures (e.g., works on the advantage of deep networks over shallow ones, see the cited papers of Montúfar et al. 2014 and Cohen et al. 2016). Our paper aims to lay the groundwork for understanding allocation strategies in a controlled setting, and even with these control settings we find surprising results, such as the sharp transition to maximal allocations in large networks; future work could explore how these allocation insights transfer to broader tasks, as mentioned in our discussion of future research directions.
>
> **Comment**:  *...in a very specific setting, limiting the scope and potential insights that could be obtained from this work, "...where the student and the teacher have matching architectures, making it difficult to extend the results to real-world applications even heuristically."  "The numerical results shown in this paper are also limited to this synthetic task only.*
>
> **Response**: We respectfully disagree with the assertion that our framework is overly restricted or difficult to extend. The student-teacher setup is a well-established framework for studying machine learning problems in controlled settings. It has been widely used in the literature to analyze generalization, expressivity, and optimization (e.g., Saglietti et al. NeurIPS 2022, which also includes a comprehensive set of references).
>
> The choice of a student-teacher setup in our work is deliberate and is done for clarity of the paper, as it isolates the reduction in expressive power arising solely from the allocation of learnable weights, rather than confounding factors such as differences in architecture or neuron nonlinearities. Specifically, when the teacher and student share the same architecture, any decrease in the student's expressive power is attributable solely to the restriction in learned weights and their allocation. This allows us to rigorously estimate the approximation error stemming from allocation strategies, independent of other factors that might limit the student’s ability to fit the labeled data.
>
> While we used the student-teacher setup for clarity, the framework is not inherently limited to this context. In fact, the teacher and student could differ in architecture, and the analysis could extend to general labeled data (our proofs remain valid under these conditions).
>
> Importantly, the MP metric itself provides a general and intuitive measure of approximation error—the reduction in expressive power due to different allocation strategies. By design, MP quantifies the probability that a model matches given labels (either iid-drawn or generated by a teacher). This aligns closely with classical definitions of expressivity, such as those introduced by the classical work of Cover (1965) in the context of perceptrons. Unlike alternative metrics in the literature, such as universality or expressive efficiency, we believe that MP offers a more interpretable and direct measure of a model's capacity.
>
> Finally, the goal of presenting the numerical results on the synthetic task is to demonstrate the validity of our theoretical analysis and to extend it to cases where the MP cannot be rigorously calculated within this task. To further demonstrate the flexibility of our approach to structured data, and without relying on a teacher, we now include new empirical results (Figure 3c) where the match probability (MP) is estimated on MNIST data using its true labels, without any teacher involved. These results illustrate that our framework is applicable to real-world datasets and is not confined to synthetic data or teacher-generated labels. We also revised the setting and discussion sections accordingly.

---

> ### Author Response · Authors · 2024-11-19
> **Detailed reply to reviewer mi9c (2/3)**
>
> **Comment**: *Various assumptions are dispersed and show up in various places within the text.*
>
> **Response**: Thank you for the comment. We have consolidated the assumptions by creating an "Assumptions" paragraph within the settings section to improve clarity. Specifically, we moved the assumption about the input distribution creating invertible matrices, previously found on line 197, to this new section. Additionally, we reviewed the manuscript to identify and streamline any remaining assumptions dispersed throughout the text.
>
> **Comment**: *Multiple proofs in this paper are written in a very informal way.*
>
> **Response**: We reviewed the entire paper carefully and improved the rigor of multiple proofs in our revision. As noted by the reviewer, some parts of Proof B1, including the statement on eigenvectors as spaces, were unclear. We revised this section to clarify and enhance its formal presentation. We also improved the clarity and rigor of proofs in Theorems B6 and B9 and would welcome any further suggestions the reviewer may have after reading the revised manuscript.
>
> **Comment**: *Have you considered correlated weights in the teacher, or introducing structures into the teacher weights?*
>
> **Response**: As stated above, most of our proofs do not rely on the presence of a "teacher" model, and thus correlated weights in the teacher are not that important. Instead, our setting applies generally to a given set of data and labels. Thus, following the reviewer's comment, in principle, it will be interesting to explore if our insights pertain to structured data.
>
> In most proofs (specifically outside Theorems 3.2, and 4.1), we only assumed invertibility and independence conditions for the data, rather than iid Gaussian distributions. To examine the effects of structured data, that goes beyond our theory, we added experiments using MNIST data, where the data contain correlations within each example rather than pure iid distributions. The fact that it is better to distribute the fixed number of learnable weights throughout the network also applies under these conditions.
>
> **Comment**: *Is this consistent with existing findings about learning in the brain?*
>
> **Response**: This is an excellent question. As we noted in the introduction, little is known on the extent of learnable weights in the brain. Due to technological constraints, it has been notoriously difficult to track synaptic weight changes in real-time as animals learn new tasks. What is known, however, is that learning induces changes in neural activity that can be highly distributed both within and across brain regions. Interestingly, a recent study demonstrated that even when learning is localized to a subset of neurons and synaptic weights, the resulting activity can propagate throughout fixed-weight connections, leading to widespread changes in neural activity. Thus, it is challenging to relate the widespread neural activity observed to the extent of changes in synaptic weights in the brain.
>
> Interestingly, recent technological advances now allow researchers to monitor effective connectivity at scale, capturing the behavior of numerous synapses across extensive brain areas over time (e.g. Finkelstein et al. 2023). With these innovations, the predictions and insights from our study could become testable in the near future, and is one of the motivations for our current study. We now discuss this point in the supplementary discussion of the paper.
>
> **Comment**: *does the degree of the polynomials play a role?*
>
> **Response**: Yes, the degree plays a role. As the degree increases, the expected number of solutions tends to grow. Additionally, the variance of the solution count approaches zero as the number of equations increases, indicating concentration around the expectation. Therefore, generally, higher-degree polynomials tend to increase the expected number of solutions, which in turn raises the likelihood of obtaining at least one solution when the number of equations grows. See the cited paper of Subag (2024) for more details.
>
> In our case, we showed that the degree of the polynomial can be reduced to a fixed degree of 2 (see lemma 3.3), therefore, our primary focus was on ensuring low variance to achieve this concentration.
>
> **Comment**: *In the proof of Theorem B1, do you assume that $W$ is diagonalizable?*
>
> **Response**: Thank you for bringing this up. Indeed, we assumed in this theorem that $W$ is diagonalizable. The measure of real matrices that are not diagonalizable over the complex equals 0, so the probability for a random matrix with a continuous probability distribution to be non-diagonalizable vanishes. We clarified this in the revised proof of Theorem B1.
>
> **Comment**: *The sentence 'the eigenvector xxx is a space' does not make sense to me.*
>
> **Response**: We appreciate this note and have clarified the wording in this section to improve comprehensibility.

---

> ### Author Response · Authors · 2024-11-19
> **Detailed reply to reviewer mi9c (3/3)**
>
> **Comment**: *Furthermore, I do not see why it is important to prevent the network from being degenerate*
>
> **Response**:  We agree with the reviewer that, in principle, the teacher could be degenerate. We did not explore this case as such degeneracy implies that even a fully-learned LRNN could not express any linear function. To clarify this, we have added this assumption to the "Assumptions" paragraph within the settings section.

---

> ### Comment · Reviewer_mi9c · 2024-11-22
> **response to rebuttal**
>
> I am very glad to see that the authors make a significant effort to refine the proofs, and thereby raised my score to 3.
>
> My remaining concern for this paper is the lack of numerical experiments. I agree with the authors that certain assumptions are required for theoretical works, however, it is very important to demonstrate empirically the potential insights that the theories have in more realistic settings. I'm happy to see that the authors added an experiment with the MNIST dataset. Along this line, I believe the paper can benefit from showing more results in nonlinear networks. Currently the only result regarding nonlinear networks is one panel and the point raised seems to be only a very qualitative prediction that having the weights distributed in more rows increase the expressivity, and increasing n helps. The paper also only shows two lines without any errorbars. I don't find the results to be very insightful if that is the only conclusion that one can extend to nonlinear networks. It would be nice if the authors can show more numerical results demonstrating more quantitative predictions that their theoretical results can have, and add some discussions on the potential applications/implications of their theory.
>
> A minor point about illustration, if i understand correctly, it seems that the MP curves should extrapolate to 0 on the left side as the fraction of learned rows approach d/n? If so, can you show that explicitly in your current figures? It would be nice to show in the figures also where your theoretical prediction of the transition is, i believe this would make the figures more self-explanatory.

---

> > ### Author Response · Authors · 2024-11-24
> > **Detailed response to reviewer mi9c**
> >
> > We appreciate the reviewer’s comment and were glad to see that they appreciate our efforts to improve the manuscript along with their comments. While the current version of the manuscript includes only one panel on nonlinear networks (in fact, two panels in the revised manuscript, which includes MNIST), we want to emphasize that even though they lacked a panel, the theoretical results we present for nonlinear networks should not be overlooked. These results lay the groundwork for developing a theory on allocations in ReLU networks, which could potentially extend to deep networks. For instance, one important quantitative prediction that arises from the theory is the existence of minimal allocations even in the nonlinear case (see line 462 in the revised version).
> >
> > As mentioned, our paper is primarily theoretical, and we believe it provides significant insights into a novel question on the behavior of allocations in neural networks, such as the sharp transition from minimal to maximal allocations in large networks. We believe that these results are non-trivial and somewhat surprising, even without combining them with numerical experiments on complex networks and real-world data. However, given the encouraging success of our theoretical predictions with the MNIST dataset, we plan to include additional experiments. Due to the constraints of the discussion phase, we are unable to incorporate additional experiments by the end of the discussion phase, but will add them to the camera-ready version to further support these findings. Similarly, regarding the errorbars and the illustration point, we thank the reviewer for these comments and will incorporate them.

---

### Official Review · Reviewer_yDsn · 2024-11-04

**Soundness:** 3
**Presentation:** 3
**Contribution:** 1
**Rating:** 3
**Confidence:** 4

**Summary:**

The paper investigates the conditions under which 'allocations' have maximal or minimal expressive power in linear RNN and linear MLPs.

**Strengths:**

Originality: I think allocation is an interesting concept though it seems to significantly overlap with the concept of subnetwork in lottery ticket hypothesis.

Clarity: The paper is clearly written.

**Weaknesses:**

Quality: Studying expressivity of RNNs/MLPs from the perspective of linear solvable equations require strong assumptions such as 'the distribution $\mathcal X$ is such that any drawn square matrix is invertible' (l197) and I'm having trouble extracting insights from the theorems.

Significance: Increasing the percentage of learnable weights leads to more expressivity seems trivial to me (maybe there are insights I'm not seeing from the paper?) so I'm not quite sure what we can learn from the submission.

**Questions:**

a. Could the authors comment on the practical implications of the theorems?

b. L147 is joint not conditional? If so notation is confusing.

---

> ### Author Response · Authors · 2024-11-19
> **Detailed reply to reviewer yDsn**
>
> We thank the reviewer for their comments. We went over the points raised by the reviewer and addressed them as detailed in the following text.
>
> **Comment**: *Increasing the percentage of learnable weights leads to more expressivity seems trivial to me (maybe there are insights I'm not seeing from the paper?) so I'm not quite sure what we can learn from the submission.*
>
> **Response**: We agree with the reviewer that increasing the percentage of learnable weights leads to more expressivity. This is NOT the setup of the paper. To avoid this naive setting, we examined the case where the number of learnable weights is fixed and explored how their allocation within the network impacts expressivity. This is stated in the abstract: ”...how different allocations of a **fixed number of learnable weights** influence the capacity of neural networks”. What we find is that by varying the distribution of these fixed numbers of learnable weights across the network (what we call learning under ‘resource constraints’), we observe substantial differences in expressivity that are nontrivial and often surprising. We now clarify this point in several places to avoid such crucial misunderstanding. For example, we now write: “distributing the *same* number of learnable weights over more neurons, such that there are fewer learnable weights per neuron, increases the network’s expressive power.”
>
> **Comment**: *The concept seems to significantly overlap with the subnetwork idea in the lottery ticket hypothesis.*
>
> **Response**: While we understand the apparent similarity, and indeed we included that in the related work of our paper, our work fundamentally differs from the lottery ticket hypothesis in both motivation and approach. The lottery ticket hypothesis focuses on identifying the existence of a subnetwork within an over-parameterized network that, when trained in isolation, achieves similar performance to the original network. In contrast, our work specifically addresses how to optimally allocate a fixed number of weights across a network to maximize expressivity. Our focus is not merely on identifying subnetworks but on understanding and optimizing allocation strategies for learnable weights within the network. We now include this point in the supplementary discussion.
>
> **Comment**:  *Studying expressivity of RNNs/MLPs from the perspective of linear solvable equations require strong assumptions such as 'the distribution X is such that any drawn square matrix is invertible'*
>
> **Response**: First, it should be noted that in the more interesting cases we study, of allocations in the recurrent weights of RNNs or when allocations are distributed across layers, the problem becomes **non-linear** and does not depend on the ability to invert matrices. The framework we propose, which reduces the nonlinear problem of recurrent weight allocation to a set of quadratic polynomials (Lemma 3.3), goes far beyond the perspective of studying RNNs through linear solvability.
>
> Second, to ensure rigor in theoretical results, certain assumptions are essential. We aim to keep these assumptions as minimal as possible while still allowing for meaningful conclusions. We now clarify our assumptions in the setting paragraph. Regarding the invertibility of matrices, in real-world problems matrices are typically invertible, and thus we do not think that these assumptions are too strong.
>
> **Comment**: *I’m having trouble extracting insights from the theorems.*
>
> **Response**: The primary insight from our work is that with a fixed number of allocated weights, distributing these weights optimally across the network can lead to maximal expressivity. We also identify specific patterns within minimal and maximal expressivity that reveal unexpected phenomena—particularly the sharp transitions in expressivity based on allocation patterns. Furthermore, to connect theory with practice, we have conducted experiments on the MNIST dataset, where more realistic input structures are used.
>
> **Comment**:  *L147 is joint not conditional? If so notation is confusing.*
>
> **Response**: Thank you for this comment. This is not conditional. We updated the notation to avoid any ambiguity.

---

### Author Response · Authors · 2024-11-19
**General response to the review**

We thank the reviewers for their thoughtful and detailed comments. We recently uploaded a revised version, incorporating the reviewers’ comments. Detailed rebuttals are posted as a reply to each reviewer. Here we review the key concerns, focusing on those that were either shared among reviewers or particularly central to the review, and summarize the main changes in the revised version.

As pointed out by the reviewers, the question of how to allocate a subset of learnable parameters in a neural network is novel. While this topic relates to existing concepts such as the lottery ticket hypothesis or continual learning, our work addresses it from a fundamentally different perspective. One of our main findings is that distributing learnable weights across the network results in significantly better expressivity. We emphasize this because it appears to have been misunderstood in the feedback of one of the reviewers (yDsn) that commented that the fact that increasing the number of learnable weights leads to greater expressivity is trivial. We agree with the reviewer. Unfortunately, they missed the central point of our work: our analysis concerns a *fixed* number of learnable weights. This distinction is crucial and it is now clarified more explicitly in the revised version to prevent any potential misunderstanding for future readers.

Additionally, our work is primarily theoretical. Theoretical analysis necessarily involves assumptions. To validate the theoretical findings, we conducted numerical simulations that adhere to these assumptions. While the purpose of these simulations was not to demonstrate real-world applicability, we have now extended our analysis to include match probability estimates for allocations on models trained with MNIST data. These results suggest that the insights derived from our theoretical work can extend to structured, real-world data.

Furthermore, while one reviewer suggested that our expressivity metric is overly restrictive (mi9c), we respectfully disagree. Match probability (MP) is a general and intuitive measure of expressivity that can be applied across a wide range of models, including more complex architectures and real-world datasets. Its utility as a benchmark was also acknowledged by another reviewer (HB5t). In response to this feedback, we provided a detailed explanation of the strengths and general applicability of MP and clarified this point in the revised manuscript to emphasize its broader relevance and significance.

Finally, we made several clarifications and revisions to the manuscript. First, we emphasized that the study focused on the effect of allocating a **fixed number of learnable weights** and differentiated our approach from the lottery ticket hypothesis. Connections to biological findings were expanded in the supplementary discussion. We consolidated all dispersed assumptions into a dedicated "Assumptions" section and refined and enhanced the rigor of proofs.

---

### Meta-Review · Area_Chair_GZt9 · 2024-12-16

**Metareview:**

The paper considers a teacher-student learning framework, where the teacher is tasked with replicating as best as possible the outputs of the teacher while training only a subset (in percentage) of the weights of a model with a similar architecture. The authors investigate the optimal solution in several models, including linear recurrent models and shallow ReLU networks.

The paper had a mixed set of reviews, ranging from definite accept to definite reject. Three reviewers interacted during the rebuttal, although the scores remained consistent. As I detail below, of the four reviews I have decided to ignore one which (by their own admission) mischaracterized the setup of the work. Of the other three reviewers, one is excited about the potential, one recommends acceptance with some remaining concerns, and one argues for rejection.

I note that the technical validity of the paper has not been questioned by the reviewers (one reviewer claimed some proofs were "informal" but did not provide examples, and I could not find them in the paper myself). Criticisms concern possible limited scope, and no direct application to "real-world" setups. On the first point, I agree with the author's rebuttal (see below), and I believe this paper has the potential to generate novel research on the field. I also agree that theoretical papers should strive for correctness but should not necessarily "beat SOTA". As a result, I lean towards acceptance in light of the possible impact and technical correctness.

**Additional Comments On Reviewer Discussion:**

- **Reviewer HB5t** was highly enthusiastic about the work, with some minor concerns that were addressed in the rebuttal. While they also highlight the fundamentally theoretical setup of the paper, they claim to be "excited about this work".

- **Reviewer QyMh** also commented that the paper is well written and the results intriguing. They are mostly concerned about (a) the fact the gradient descent does not perform well in this context, and (b) about the limited size of the experiments. On my side, I believe both points are valid directions for future work but they do not detract from the validity of the paper, especially as the subset allocation task is challenging from a GD optimization perspective.

- **Reviewer mi9c** confirmed the validity of the work but had two major concerns: (a) limited scope of the setup, and (b) no numerical experiments. They gave an extremely low score (3) as a result. For point (a), the authors have argued that controlled student-teacher models are common to formulate similar scenarios, while for point (b) they have added some experiments on MNIST but claimed the paper is mostly theoretical. After careful evaluation I agree with the authors on both points. I don't think theoretical papers should necessarily "extend the results to real-world applications even heuristically", as the impact and validity of a theoretical paper can easily be delayed as the theory is refined and applied.

- **Reviewer yDsn** made a very short review claiming "increasing the percentage of learnable weights leads to more expressivity seems trivial to me". I agree with the authors that this is a mischaracterization of the work. As a result (and since the reviewer has not interacted during rebuttal, and the original review was clearly wrote in haste) I have ignored this review for my final decision.

---

### Decision · Program_Chairs · 2025-01-22

Accept (Poster)